# Supervised Quadratic Feature Analysis: Information Geometry for Dimensionality Reduction

**Daniel Herrera-Esposito**                                          *dherresp@sas.upenn.edu*
*Department of Psychology*
*University of Pennsylvania*

**Johannes Burge**                                                  *jburge@psych.upenn.edu*
*Department of Psychology*
*University of Pennsylvania*

**Reviewed on OpenReview:** *https://openreview.net/forum?id=jwNJiLphnZ*

## Abstract

Supervised dimensionality reduction maps labeled data into a low-dimensional feature space while preserving class separation. A common strategy is to learn features that maximize a measure of statistical dissimilarity between the class-conditional probability distributions. Information geometry, which is rooted in Riemannian geometry, provides an alternative framework for measuring class dissimilarity. It treats probability distributions as points in a statistical manifold and uses the Fisher information metric to define a geodesic distance–the Fisher-Rao distance–between distributions The Fisher-Rao distance is an appealing candidate for measuring class separation because the Fisher information metric is a local measure of discriminability, and because it allows a geometric interpretation. Here, we present Supervised Quadratic Feature Analysis (SQFA), a supervised dimensionality reduction method which learns linear features that maximize Fisher-Rao distances between class-conditional distributions, under Gaussian assumptions. In multiple real world datasets, we find that SQFA features support classification accuracy that is competitive with features that maximize more popular measures of dissimilarity, or that are learned by other state-of-the-art dimensionality reduction methods. Notably, the best classification accuracy is achieved by SQFA-H features, a variant of SQFA that maximizes the Hellinger distance, a rarely used objective for dimensionality reduction. These results demonstrate the potential of information geometry as a tool for supervised dimensionality reduction. We provide a Python implementation of SQFA at `https://github.com/dherrera1911/sqfa`.

## 1 Introduction

Consider a random vector $\mathbf{x} \in \mathbb{R}^n$ with label $y \in \{1, \ldots, c\}$, where $c$ is the number of classes. Supervised dimensionality reduction aims to map the high-dimensional variable $\mathbf{x}$ to a lower-dimensional variable $\mathbf{z} \in \mathbb{R}^m$ that best supports classification performance, or class separation. There are many methods for learning nonlinear features, like deep neural networks (LeCun et al., 2015) and manifold learning (Sainburg et al., 2021). However, methods for learning linear features of the form $\mathbf{z} = \mathbf{F}^T\mathbf{x}$ are still in demand because of their simplicity, interpretability, and data efficiency (Cunningham & Ghahramani, 2015). A common approach to linear supervised dimensionality reduction is to learn features that maximize a statistical or information-theoretic measure of dissimilarity between pairwise class distributions (Fukunaga, 1990), like the squared Mahalanobis distance (i.e. Linear Discriminant Analysis; LDA), the Bhattacharyya distance (Choi & Lee, 2003), or the KL divergence (Tao et al., 2007; Dwivedi et al., 2022).

Information geometry provides an alternative approach for measuring the dissimilarity between probability distributions, by considering them as points in a statistical manifold. It uses the Fisher information metric, a

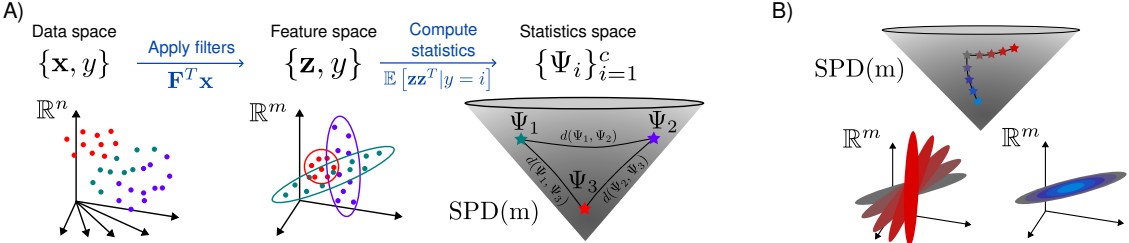

Figure 1: SQFA learns features using information geometry. **A)** SQFA and smSQFA map the $n$-dimensional data into an $m$-dimensional feature space using the linear filters $\mathbf{F}$. In smSQFA, the class-specific second-moment matrices of the features are represented as points in the SPD(m) manifold. Learning is achieved by maximizing the Fisher-Rao distances between classes in SPD(m). **B)** Each point in SPD(m) (top) corresponds to a second-moment ellipse (below). As the distance in SPD(m) increases, the second-order statistics become more different and the distributions more discriminable.

principled measure of local discriminability[1] commonly used in machine learning and neuroscience (Dayan & Abbott, 2005; Amari, 2016), to measure distances in the manifold of probability distributions. The geodesic distance induced by this metric is the Fisher-Rao distance.

Maximizing Fisher-Rao distances between classes is therefore an attractive learning objective for several reasons. First, there has been a recent surge of interest in the geometry of neural representations in both machine learning and neuroscience (Kriegeskorte & Wei, 2021; Chung & Abbott, 2021). The Fisher information metric is commonly used for characterizing local representation geometry, but it is limited to infinitesimally close distributions (Wang & Ponce, 2021; Ding et al., 2023; Feather et al., 2024; Ye & Wessel, 2024; Zhou et al., 2024). The Fisher-Rao distance provides a tool for studying the non-local geometry of neural representations (Kriegeskorte & Wei, 2021). Second, a long-standing hypothesis from perception science dating back to Riemann himself proposes that geodesic distances induced by perceptual discriminability metrics (analogous to the Fisher information metric) can capture supra-threshold (i.e. non-local) perceptual judgments of similarity (Riemann, 1867; Fechner, 1860; Vacher & Mamassian, 2024; Zhou et al., 2024; Hong et al., 2025). This makes the Fisher-Rao distance an appealing candidate for learning representations that maximize supra-threshold perceptual dissimilarities. Third, given the broad interest in information geometry, it is of standalone interest to understand how maximizing Fisher-Rao distances compares to other common objective functions in machine learning.

Despite its theoretical and practical appeal, very little work has studied the use of Fisher-Rao distances as an objective for learning (Miyamoto et al., 2024). To our knowledge, Fisher-Rao distances between classes have not been used for dimensionality reduction. One possible reason is that closed-form expressions for the Fisher-Rao distance are not available for general multivariate Gaussians (Nielsen, 2023). Another possible reason is that information-theoretic measures of dissimilarity (e.g. Bhattacharyya distance, KL divergence) are more directly linked to classification error, making them a more obvious choice (Fukunaga, 1990).

Here, we present Supervised Quadratic Feature Analysis (SQFA), a dimensionality reduction method that learns linear features by maximizing Fisher-Rao distances between class-conditional distributions, under Gaussian assumptions. We use an exact expression for the Fisher-Rao distance for zero-mean Gaussians, and the Calvo-Oller lower bound (Calvo & Oller, 1990) as a closed-form approximation in the general case. We compare the classification performance supported by SQFA features to that supported by features that maximize other measures of dissimilarity. We also compare performance to other popular methods for dimensionality reduction. Our analyses show that:

---

[1]We use the term discriminability loosely to refer to the ability to discriminate between two distributions based on observed samples. Local discriminability of $p(\mathbf{z}|\boldsymbol{\theta})$ in the direction $\boldsymbol{\theta}'$ refers to the ability to discriminate between $p(\mathbf{z}|\boldsymbol{\theta})$ and $p(\mathbf{z}|\boldsymbol{\theta} + \epsilon\boldsymbol{\theta}')$, where $\epsilon\boldsymbol{\theta}'$ is a small perturbation. Local discriminability is closely related to Fisher information via the Cramér-Rao bound

- Maximizing Fisher-Rao distances between classes leads to features that support classification accuracy competitive with state-of-the-art methods for QDA and kNN classifiers.

- Maximizing the Calvo-Oller bound is a practical way to maximize the Fisher-Rao distance.

- Features learned by maximizing the Hellinger distance (SQFA-H) consistently support the highest classification accuracy. SQFA-H is a state-of-the-art method for learning linear features that maximize classification accuracy.

## 2 Fisher-Rao distance as a discriminability proxy

**Notation.** The class-conditional means and covariances of the data variable $\mathbf{x}$ are denoted by $\boldsymbol{\gamma}_i = \mathbb{E}\left[\mathbf{x}|y = i\right]$ and $\boldsymbol{\Phi}_i = \mathbb{E}\left[(\mathbf{x} - \boldsymbol{\gamma}_i)(\mathbf{x} - \boldsymbol{\gamma}_i)^T|y = i\right]$, respectively. The low-dimensional projection of $\mathbf{x}$ is given by $\mathbf{z} = \mathbf{F}^T\mathbf{x}$, where $\mathbf{F} \in \mathbb{R}^{n \times m}$ is a matrix of filters, and the respective class-conditional means, second moments, and covariances are given by $\boldsymbol{\mu}_i = \mathbb{E}\left[\mathbf{z}|y = i\right]$, $\boldsymbol{\Psi}_i = \mathbb{E}\left[\mathbf{z}\mathbf{z}^T|y = i\right]$, and $\boldsymbol{\Sigma}_i = \boldsymbol{\Psi}_i - \boldsymbol{\mu}_i\boldsymbol{\mu}_i^T$. We also denote by $\boldsymbol{\theta}_i$ the parameters of $p(\mathbf{z}|y = i) = p(\mathbf{z}|\boldsymbol{\theta}_i)$, which in the Gaussian cases is $\boldsymbol{\theta}_i = (\boldsymbol{\mu}_i, \boldsymbol{\Sigma}_i)$.

### 2.1 Supervised dimensionality reduction via dissimilarity maximization

The goal of linear supervised dimensionality reduction is to find filters $\mathbf{F}$ such that the classes are as discriminable as possible in the low-dimensional space of the variable $\mathbf{z} = \mathbf{F}^T\mathbf{x}$. A common way to achieve this is to maximize a dissimilarity measure between the class-conditional distributions. For Gaussian-distributed classes $p(\mathbf{z}|y = i) = \mathcal{N}(\boldsymbol{\mu}_i, \boldsymbol{\Sigma}_i)$, given a dissimilarity measure $d(\boldsymbol{\theta}_i, \boldsymbol{\theta}_j)$ where $\boldsymbol{\theta}_i = (\boldsymbol{\mu}_i, \boldsymbol{\Sigma}_i)$, the problem reduces to

$$\underset{\mathbf{F} \in \mathbb{R}^{n \times m}}{\arg\max} \sum_{i=1}^{c} \sum_{j=1}^{c} d(\boldsymbol{\theta}_i, \boldsymbol{\theta}_j) \tag{1}$$

If $d(\boldsymbol{\theta}_i, \boldsymbol{\theta}_j)$ is a good proxy for discriminability, the learned features should support high classification accuracy (Figure 1). We argue that the Fisher-Rao distance, $d_{FR}(\boldsymbol{\theta}_i, \boldsymbol{\theta}_j)$, is a sensible proxy for discriminability.

### 2.2 Fisher-Rao distance as accumulated local discriminability

For a distribution $p(\mathbf{z}|\boldsymbol{\theta})$, the Fisher information in the direction $\boldsymbol{\theta}'$ in the parameter space is defined as

$$\mathcal{I}_{\boldsymbol{\theta}}(\boldsymbol{\theta}') = \mathbb{E}\left[(s(\mathbf{z}, \boldsymbol{\theta}) \cdot \boldsymbol{\theta}')^2\right] = \boldsymbol{\theta}'^T \mathbf{J}(\boldsymbol{\theta})\boldsymbol{\theta}' \tag{2}$$

where the expectation is over $\mathbf{z}$, $s(\mathbf{z}, \boldsymbol{\theta}) = \nabla_{\boldsymbol{\theta}} \log p(\mathbf{z}|\boldsymbol{\theta})$ is the score function, and $\mathbf{J}(\boldsymbol{\theta}) = \mathbb{E}[s(\mathbf{z}, \boldsymbol{\theta})s(\mathbf{z}, \boldsymbol{\theta})^T]$ is the Fisher information matrix. The quantity $\sqrt{\mathcal{I}_{\boldsymbol{\theta}}(\boldsymbol{\theta}')}$ measures the discriminability between $p(\mathbf{z}|\boldsymbol{\theta})$ and $p(\mathbf{z}|\boldsymbol{\theta} + \epsilon\boldsymbol{\theta}')$, where $\epsilon\boldsymbol{\theta}'$ is a small perturbation.

Let $\boldsymbol{\theta}(t) = (\boldsymbol{\mu}(t), \boldsymbol{\Sigma}(t))$ be the Fisher-Rao geodesic (i.e. the shortest curve) connecting $\mathcal{N}(\boldsymbol{\mu}_i, \boldsymbol{\Sigma}_i)$ to $\mathcal{N}(\boldsymbol{\mu}_j, \boldsymbol{\Sigma}_j)$ along the manifold of Gaussian distributions, where $\boldsymbol{\theta}(0) = \boldsymbol{\theta}_i$, $\boldsymbol{\theta}(1) = \boldsymbol{\theta}_j$, and $\boldsymbol{\theta}'(t)$ is the velocity. The Fisher-Rao distance is obtained by integrating the speed along the geodesic, $d_{FR}(\boldsymbol{\theta}_i, \boldsymbol{\theta}_j) = \int_0^1 \|\boldsymbol{\theta}'(t)\|dt$. The Fisher information metric defines the speed as $\|\boldsymbol{\theta}'\| = \sqrt{\mathcal{I}_{\boldsymbol{\theta}}(\boldsymbol{\theta}')}$, which for the Gaussian case is

$$\|\boldsymbol{\theta}'(t)\| = \sqrt{\mathcal{I}_{\boldsymbol{\theta}(t)}(\boldsymbol{\theta}'(t))} = \left[\boldsymbol{\mu}'(t)^T\boldsymbol{\Sigma}(t)^{-1}\boldsymbol{\mu}'(t) + \tfrac{1}{2}\operatorname{Tr}\left(\boldsymbol{\Sigma}(t)^{-1}\boldsymbol{\Sigma}'(t)\boldsymbol{\Sigma}(t)^{-1}\boldsymbol{\Sigma}'(t)\right)\right]^{1/2} \tag{3}$$

Then, $\|\boldsymbol{\theta}'(t)\|$ is a measure of the discriminability between $\mathcal{N}(\boldsymbol{\mu}(t), \boldsymbol{\Sigma}(t))$ and $\mathcal{N}(\boldsymbol{\mu}(t + \mathrm{d}t), \boldsymbol{\Sigma}(t + \mathrm{d}t))$. Thus, the Fisher-Rao distance can be expressed as $d_{FR}(\boldsymbol{\theta}_i, \boldsymbol{\theta}_j) = \int_0^1 \sqrt{\mathcal{I}_{\boldsymbol{\theta}(t)}(\boldsymbol{\theta}'(t))}dt$, and it can be conceptualized as the accumulated discriminability of the infinitesimal perturbations transforming $\mathcal{N}(\boldsymbol{\mu}_i, \boldsymbol{\Sigma}_i)$ into $\mathcal{N}(\boldsymbol{\mu}_j, \boldsymbol{\Sigma}_j)$ along the geodesic, making it a sensible proxy for discriminability.

### 2.3 Closed-form expressions for the Fisher-Rao distance

There is no closed-form expression for the Fisher-Rao distance between arbitrary Gaussians. Numerical methods exist to compute it (Nielsen, 2023; Nielsen & Soen, 2024), but they are too costly to be used as

optimization objectives. To overcome this problem, we use two complementary approaches: 1) we consider the special case of zero-mean Gaussians, and 2) we use a closed-form lower bound for the general case.

**Zero-mean Gaussians.** The manifold of $m$-dimensional zero-mean Gaussians is equivalent to the manifold of $m$-by-$m$ symmetric positive definite matrices, SPD(m), where each point corresponds to a covariance matrix. The Fisher-Rao distance between zero-mean Gaussians, denoted $d_{FR}(\mathbf{\Sigma}_i, \mathbf{\Sigma}_j)$, is proportional to the affine-invariant distance $d_{AI}$ in SPD(m) (Atkinson & Mitchell, 1981)

$$d_{AI}(\mathbf{\Sigma}_i, \mathbf{\Sigma}_j) = \sqrt{\sum_{k=1}^{m} \log^2 \lambda_k} = d_{FR}(\mathbf{\Sigma}_i, \mathbf{\Sigma}_j)\sqrt{2} \tag{4}$$

where $\lambda_k$ is the $k$-th generalized eigenvalue of the pair $(\mathbf{\Sigma}_i, \mathbf{\Sigma}_j)$. Leveraging this closed-form expression, we define smSQFA (for 'second-moment SQFA'), which assumes zero-mean Gaussians, and that maximizes pairwise $d_{AI}(\mathbf{\Psi}_i, \mathbf{\Psi}_j)$, where $\mathbf{\Psi}_i$ is the second-moment matrix for class $i$ in the feature space.

Zero-mean distributions are relevant for some applications, like EEG recordings (Horev et al., 2017), some local visual tasks (Burge, 2020), and auditory waveform data, among others. Also, the zero-mean case is more tractable and allows one to draw direct links between the Fisher-Rao distance and measures of discriminability, such as the ratio between class-conditional feature variances (see Appendix A). However, our main motivation for considering the zero-mean case is that it allows us to maximize the exact Fisher-Rao distance, which is important for understanding the link between maximizing Fisher-Rao distances and discriminability without the confound of using an approximation (see below).

**Calvo-Oller bound.** For arbitrary Gaussians, we use the Calvo-Oller bound (Calvo & Oller, 1990). It is obtained by embedding $\boldsymbol{\theta}$ into SPD(m + 1) as

$$\mathbf{\Omega} = \begin{bmatrix} \mathbf{\Sigma} + \boldsymbol{\mu}\boldsymbol{\mu}^T & \boldsymbol{\mu} \\ \boldsymbol{\mu}^T & 1 \end{bmatrix} \tag{5}$$

The Calvo-Oller formula is given by $d_{CO}(\boldsymbol{\theta}_i, \boldsymbol{\theta}_j) = d_{AI}(\mathbf{\Omega}_i, \mathbf{\Omega}_j)/\sqrt{2}$, and it provides a lower bound for $d_{FR}(\boldsymbol{\theta}_i, \boldsymbol{\theta}_j)$. This bound has desirable properties: it is a true distance, it matches $d_{FR}$ when $\boldsymbol{\mu}_i = \boldsymbol{\mu}_j$ (Appendix B), and it is invariant to invertible linear transformations of $\mathbf{z}$ (Nielsen, 2023). SQFA learns filters by maximizing the Calvo-Oller bound between classes (Algorithm 1). Empirically, we found that the Calvo-Oller bound is a good proxy for the Fisher-Rao distance in the analyzed datasets. It is also worth noting that the Calvo-Oller bound can be easily extended to other elliptical distributions (e.g. multivariate t-Student and Cauchy distributions) (Calvo & Oller, 2002; Nielsen, 2023), which may facilitate future work that extends SQFA to non-Gaussian distributions.

## 3 Related work

### 3.1 Parametric methods maximizing dissimilarity between class-conditional distributions

Maximizing a measure of dissimilarity between class-conditional distributions is an established approach for supervised dimensionality reduction. For example, the canonical method LDA can be interpreted as maximizing pairwise squared Mahalanobis distances (Appendix C). However, the Mahalanobis distance is limited by the assumption that all class-conditional distributions have identical covariance. To overcome this limitation, other methods propose using other measures of dissimilarity. Under Gaussian assumptions the most popular are the Chernoff distance and its special case, the Bhattacharyya distance[2] (Duin & Loog, 2004; Rueda & Herrera, 2008; Choi & Lee, 2003). One reason for the success of these distances is that they provide an upper bound for the Bayes classification error (Kailath, 1967; Fukunaga, 1990). Another reason is that they have simple closed-form expressions.

---

[2]Neither the Bhattacharyya nor the Chernoff distances are a true distance, since they do not satisfy the triangle inequality. Nonetheless, we refer to them as distances as is common in the literature.

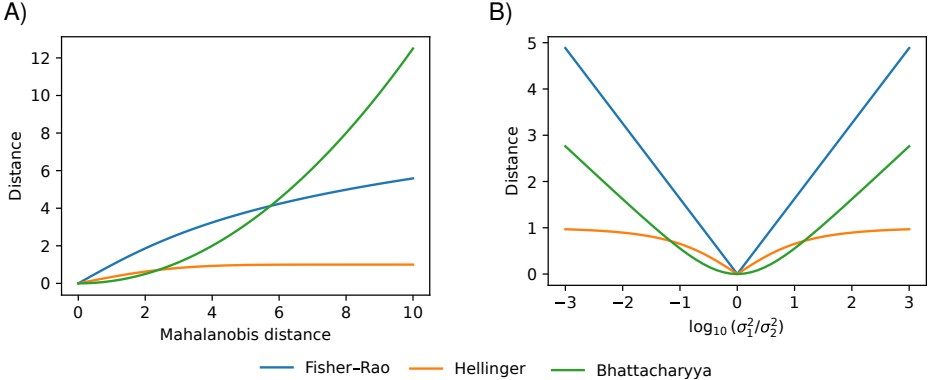

Figure 2: Growth in Fisher-Rao, Hellinger, and Bhattacharyya distances for special cases with closed-form expressions. **A)** Distances between Gaussians with identical covariance but different means (the Mahalanobis distance is proportional to the distance between the means along a line). **B)** Distance between Gaussians in 1D with identical means but different variances.

The Bhattacharyya distance between two distributions $p(\mathbf{z})$ and $q(\mathbf{z})$ is given by $d_B(p, q) = -\log BC(p, q)$, where $BC(p, q) = \int \sqrt{p(\mathbf{z})q(\mathbf{z})}d\mathbf{z}$ is the dot product between the square-root densities. For two Gaussians with parameters $\boldsymbol{\theta}_i$ and $\boldsymbol{\theta}_j$, denoting $\boldsymbol{\Sigma} = (\boldsymbol{\Sigma}_i + \boldsymbol{\Sigma}_j)/2$, the distance is given by

$$d_B(\boldsymbol{\theta}_i, \boldsymbol{\theta}_j) = \frac{1}{8}(\boldsymbol{\mu}_i - \boldsymbol{\mu}_j)^T \boldsymbol{\Sigma}^{-1}(\boldsymbol{\mu}_i - \boldsymbol{\mu}_j) + \frac{1}{2}\log\frac{\det(\boldsymbol{\Sigma})}{\sqrt{\det(\boldsymbol{\Sigma}_i)\det(\boldsymbol{\Sigma}_j)}} \tag{6}$$

Another important measure of dissimilarity is the Hellinger distance, defined as the Euclidean distance between square-root densities, $d_H(p, q) = \sqrt{\frac{1}{2}\int(\sqrt{p(\mathbf{z})} - \sqrt{q(\mathbf{z})})^2 d\mathbf{z}}$. It relates to the Bhattacharyya distance via $d_H(p, q) = \sqrt{1 - e^{-d_B(p,q)}}$. Therefore, it also has a closed-form expression for Gaussians, and provides the same bound on the Bayes classification error. It is a true distance, bounded between 0 and 1, and it converges to the Fisher-Rao distance for infinitesimally close distributions (Amari, 2016). Notably, $d_H$ has not been used for dimensionality reduction in the multiclass Gaussian case (see Carter et al. (2009) for a non-parametric example, and Dwivedi et al. (2022) for a two-class Gaussian example).

Unlike the Fisher-Rao distance, $d_B$, and $d_H$ are not geodesic distances along a statistical manifold. However, as mentioned, an appealing property of $d_B$ and $d_H$ for dimensionality reduction is that they provide bounds on the Bayes classification error between two classes (Fukunaga, 1990). On the other hand, the Fisher-Rao distance on the manifold of Gaussian distributions does not have a direct link to Bayes error. But while this seemingly favors $d_B$ and $d_H$ as learning objectives, the relation to Bayes error in the two-class case does not always translate well to the multiclass case (Loog et al., 2001; Thangavelu & Raich, 2008). For the multiclass case, the objective function needs to balance trade-offs between increases in different pairwise distances. Different distances can place different relative weights on the class pairs, based on how the distance grows as two classes become further apart (Figure 2). This can lead two different distances to favor different solutions, even if they are equivalent for the two-class case. For example, Figure 2A shows that two pairs of classes with Mahalanobis distances of 1 and 8, contribute to the Bhattacharyya, Fisher-Rao, and Hellinger distance objectives with relative weights of 1:64, 1:5, and 1:3 respectively. The consequences of this are illustrated by a toy example in Appendix D.1, where the Fisher-Rao distance displays higher robustness to outliers than the Bhattacharyya distance. Therefore, for the multiclass problem it is largely an empirical question which objective is better for a given dataset and a given goal.

### 3.2 Non-parametric methods based on pairwise sample distances

Metric learning is another popular family of methods with the goal of maximizing the distance between data points from different classes, while minimizing the distance between data points from the same class. These methods do not assume parametric distributions.

A classic example is Local Fisher Discriminant Analysis (LFDA) (Sugiyama & Jp, 2007), which aims to maximize the ratio of between-class and within-class scatters like LDA, but it uses local scatter matrices obtained by weighting pairs of data points by their proximity. This method is useful when the classes are multimodal, because it does not force different modes of the same class to be close together, as LDA does. Another popular method that supports high classification accuracy is Large Margin Nearest Neighbors (LMNN) (Weinberger & Saul, 2009). This method learns a subspace that maximizes the performance of the kNN classifier by bringing the $k$ nearest neighbors of the same class closer, while trying to push different classes apart by a large margin.

Another interesting method with a geometric motivation is Wasserstein Discriminant Analysis (WDA) (Flamary et al., 2018). WDA tries to maximize the ratio of between-class and within-class spread, but it replaces the scatter matrices with regularized Wasserstein distances between class-conditional empirical distributions. A regularization hyperparameter controls the trade-off between global and local structure, with LDA being the special case when the regularization approaches 0. Unlike the Fisher-Rao distance, the Wasserstein distance is not directly linked to local discriminability, and it is not invariant to invertible linear transformations.

This class of methods can capture complex patterns in the data, because they do not make distributional assumptions. However, they are often computationally expensive and scale poorly with the number of dimensions and data samples.

### 3.3 Non-parametric methods based on statistical dependence with labels

Another family of supervised dimensionality reduction methods is built on the idea of maximizing statistical dependence between the low-dimensional representation of the data $\mathbf{z}$ and the labels $y$. Supervised Principal Component Analysis (SPCA) (Barshan et al., 2011) is a popular representative example. It finds the principal components that have maximal dependence on the response variable as measured using the Hilbert-Schmidt independence criterion (HSIC).

## 4 Methods

**Optimization.** Filters $\mathbf{F}$ were optimized by maximizing Equation 1 using L-BFGS, until the change in the objective was $\leq 10^{-6}$. The columns of $\mathbf{F}$ were initialized to random unit vectors and constrained to have unit norm by parametrizing them in terms of an unconstrained variable (Lezcano Casado, 2019)[3]. The class-conditional statistics of $\mathbf{z}$ were computed from the means and covariances of $\mathbf{x}$, i.e. $\boldsymbol{\mu}_i = \mathbf{F}^T \boldsymbol{\gamma}_i$ and $\boldsymbol{\Sigma}_i = \mathbf{F}^T \boldsymbol{\Phi}_i \mathbf{F}$. The optimization procedure is summarized in Algorithm 1.

Optimization is non-convex, but different random initializations generally achieved very similar performance (Appendix E). This suggests that SQFA variants are robust to the random initialization seed, and that random initialization is usually sufficient, although it might be beneficial to run multiple seeds and select the best solution.

**Regularization.** To prevent rank-deficient or ill-conditioned covariances, a regularization term was added as $\boldsymbol{\Sigma}_i = \mathbf{F}^T \boldsymbol{\Phi}_i \mathbf{F} + \mathbf{I}_m \sigma^2$, where $\mathbf{I}_m$ is the identity matrix, and $\sigma^2$ is the regularization parameter. For the digit and neural recording datasets, $\sigma^2$ was selected via grid-search using a validation set. Performance was robust across a wide range of $\sigma^2$ values (Appendix F). For the speed-estimation dataset, we set $\sigma^2 = 0.001$ to match the biologically-informed value of the original work (Burge & Geisler, 2015).

---

[3]Using an orthogonal constraint did not improve performance or training time. Additionally, orthogonalization can be performed after learning, without affecting QDA accuracy.

---

**Algorithm 1** SQFA optimization

---

**Input:** $\{\gamma_i, \Phi_i\}_{i=1}^c$, $m$, $\sigma^2$.  **Output:** $\mathbf{F} \in \mathbb{R}^{n \times m}$.

1. Initialize $\eta \in \mathbb{R}^{n \times m}$, $\mathcal{L}_{\text{prev}} \leftarrow -\infty$, $\Delta \leftarrow \infty$.

2. While $\Delta \geq 10^{-6}$:

   (a) $\mathbf{F} \leftarrow g(\eta)$, where $g$ normalizes the columns of $\eta$.

   (b) For $i = 1, \ldots, c$:
   $$\boldsymbol{\mu}_i \leftarrow \mathbf{F}^T \boldsymbol{\gamma}_i,$$
   $$\boldsymbol{\Sigma}_i \leftarrow \mathbf{F}^T \boldsymbol{\Phi}_i \mathbf{F} + \sigma^2 I_m,$$
   $$\boldsymbol{\Omega}_i \leftarrow \begin{bmatrix} \boldsymbol{\Sigma}_i + \boldsymbol{\mu}_i \boldsymbol{\mu}_i^T & \boldsymbol{\mu}_i \\ \boldsymbol{\mu}_i^T & 1 \end{bmatrix}$$

   (c) $\mathcal{L}(\eta) \leftarrow \sum_{i<j} d_{AI}(\boldsymbol{\Omega}_i, \boldsymbol{\Omega}_j)$

   (d) $\eta \leftarrow \text{L-BFGS}(\eta, \mathcal{L})$

   (e) $\Delta \leftarrow |\mathcal{L}(\eta) - \mathcal{L}_{\text{prev}}|; \quad \mathcal{L}_{\text{prev}} \leftarrow \mathcal{L}(\eta)$.

3. Return $\mathbf{F} = g(\eta)$.

---

**Complexity and scalability.** Each optimization step has complexity $O(m^3 c^2 + cmn^2)$, where $c$ is the number of classes, $m$ is the number of filters, and $n$ is the dimensionality of the data (Appendix G.1). The most expensive computations depend on the feature space dimension $m$, which is typically small. A scaling analysis on synthetic data showed that SQFA can learn up to 500 filters, and handle data with up to 20,000 dimensions on a consumer laptop over a few minutes (Appendix G.2). SQFA was among the fastest methods to train in our analyses, usually converging in a few seconds (Appendix G.3).

**Invariance.** The Fisher-Rao distance and the Calvo-Oller bound are invariant to invertible linear transformations. This implies that learned filters are unique only up to the subspace they span (Appendix I). When regularization is used, however, the invariance no longer holds. This has some consequences for learning. First, in the absence of regularization the unit norm constraint does not affect the objective, but in the presence of regularization the constraint is needed to prevent the norm of the filters from growing indefinitely. Second, in the absence of regularization the learned filters vary considerably across initializations, but they become consistent when regularization is used. See Appendix I for further discussion.

We also note that the filters are not rank-ordered by their usefulness for classification. To obtain filters rank-ordered by usefulness, the filters can be learned sequentially in pairs: two filters are learned first, then this pair can be fixed and two more can be learned, and so on. The resulting filter pairs are ordered by how well they separate the classes.

**Evaluation.** To evaluate the usefulness of SQFA for dimensionality reduction, we compared its performance to some commonly used linear dimensionality reduction techniques: PCA, LDA, SPCA (Barshan et al., 2011), LFDA (Sugiyama & Jp, 2007), LMNN (Weinberger & Saul, 2009), and WDA (Flamary et al., 2018).

Additionally, to evaluate how the Fisher-Rao distance compares to other measures of class dissimilarity for dimensionality reduction, we used a set of SQFA variants using the same optimization procedure to maximize different dissimilarity measures. For this we used the Bhattacharyya distance (SQFA-B), the Hellinger distance (SQFA-H), the Jeffreys divergence (SQFA-J), and the Wasserstein distance (SQFA-W)[4]. Note that SQFA-B is analogous to some popular methods described in the Related Work (Choi & Lee, 2003; Duin & Loog, 2004), so it also serves as a benchmark of the literature.

---

[4]For SQFA-W, we constrained the filters to be orthonormal. This was necessary to prevent SQFA-W filters to become rank-deficient.

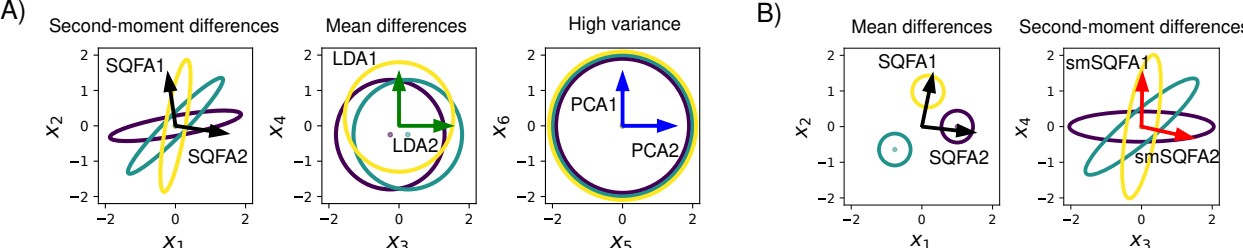

Figure 3: **A)** SQFA vs. LDA vs. PCA. Each of the three panels depicts two dimensions of a 6D data space. Ellipses show the conditional probability distributions of the data vector **x** for three classes (colors) in a 6D toy dataset. The classes are separated by different statistical properties. Classes are distinguished by large differences in the covariances (dimensions 1-2), small differences in the means (dimensions 3-4), or neither (dimensions 5-6). Two filters were learned with each of SQFA, LDA, and PCA. The learned filters are shown as arrows in the data space, indicating the axis of the data selected by each filter. SQFA prefers the most discriminative subspace. **B)** SQFA vs. smSQFA. Each of the two panels depicts two dimensions of a 6D data space. Ellipses show the conditional probability distributions of the data vector **x** for three classes (colors) in a 4D toy dataset. Classes are distinguished by large differences in the means (dimensions 1-2), and by large differences in the covariances (dimensions 3-4). We learned two filters with SQFA and smSQFA, shown as arrows in the data space. The SQFA filters select for the most discriminative subspace (dimensions 1-2).

We evaluate the different dimensionality reduction methods using QDA classification accuracy in the low-dimensional feature space. We use QDA because it is the optimal classifier under Gaussian assumptions. Since SQFA assumes Gaussian class-conditional distributions, it can be seen as optimizing QDA performance, and QDA accuracy is a natural way to evaluate SQFA features. To analyze the more general usefulness of SQFA features, we also evaluate performance using a kNN classifier in Appendix H. Remarkably, SQFA features support high accuracy even for kNN, and the conclusions are similar with both evaluation methods. Unless otherwise specified, each SQFA variant was trained with 10 different random initializations, and the results show the median across initializations. Like for SQFA, we performed a grid search to select the best regularization hyperparameters for LDA (covariance shrinkage parameter), LFDA (number of neighbors), and WDA (regularization parameter).

For LMNN and LFDA, we used the implementations from the Python package `metric-learn` (Vazelhes et al., 2020). For WDA, we used a custom PyTorch implementation following the Python package `POT` (Flamary et al., 2021)[5]. For PCA, LDA, and QDA we use the package `scikit-learn` (Pedregosa et al., 2011).

**Code.** A Python package implementing SQFA is made available at https://github.com/dherrera1911/sqfa. The code used in our analyses is available at https://github.com/dherrera1911/sqfa_analyses.

## 5 Results

### 5.1 Toy problem: SQFA vs. LDA vs. PCA

First, we illustrate the differences between SQFA and the canonical dimensionality reduction methods, LDA and PCA. For this, we designed a toy dataset with a six-dimensional variable **x** and three classes. The six-dimensional space contains three different 2D subspaces, each represented by a panel in Figure 3A. The statistics of the dataset are built such that each subspace is preferred by one of the three techniques, i.e. SQFA, LDA, or PCA. With each of the models we learned two filters on the six-dimensional dataset. The filters learned by each model are shown as arrows in the data space.

---

[5]The methods LMNN, LFDA and WDA are very computationally expensive, so we used a reduced dataset for training them. For LMNN and WDA we used 500 samples per class. For all three methods, we reduced the dimensionality down to 200 dimensions using PCA as a preprocessing step. SQFA performance was similar when using this same reduced dataset, so our conclusions are not affected by this analysis choice.

Dimensions 1-2 (Figure 3A, left) have no differences between the class means, but have very different–and hence highly discriminative–class-specific covariances. This subspace is selected for by SQFA (black arrows), because it produces the largest Fisher-Rao distances. Dimensions 3-4 (Figure 3A, center) contain slight differences between the class means, but these are not very discriminative. This subspace is selected for by LDA (green arrows) because it is the only one with differences in the means. Dimensions 5-6 (Figure 3A, right) contain large covariances, but the class-specific distributions are identical. Because this subspace contains the largest overall variances, it is favored by PCA. Filters that are learned with each of the three methods–SQFA, LDA, and PCA–select for the expected subspace.

The previous toy problem shows that SQFA can capture the differences between class-specific covariances that support classification. However, SQFA can also capture differences in the class means, and it can flexibly select one or the other depending on which one is most informative. To illustrate this, we designed a second toy dataset with three classes and a four-dimensional variable $\mathbf{x}$, composed of two 2D subspaces (like the previous example).

Dimensions 1-2 (Figure 3B, left) contain large differences in the class-specific means, supporting strong discrimination. Dimensions 3-4 (Figure 3B, right) have different class-specific covariances but identical means, supporting weaker discrimination than dimensions 1-2. We designed the classes such that their second-moment matrices (i.e. $\mathbf{\Psi}_i$) are more different for dimensions 3-4 than for dimensions 1-2. We trained both SQFA and smSQFA on the full four-dimensional dataset, and show the learned filters as arrows in the data space.

Because SQFA can select for either first-order or second-order class differences, its filters select for dimensions 1-2, where the classes are more separated (black arrows). On the other hand, smSQFA is only sensitive to class-specific second-moment matrices, so its filters select for dimensions 3-4. This shows that SQFA can flexibly select differences in either the means or the covariances.

It is worth noting that even in the case where the classes have identical covariance, SQFA can learn features that are different from those learned by LDA, because both methods maximize different pairwise distances, so they put different relative weights on the class pairs (see Section 3.1). A toy example illustrating this is shown in the Appendix D.1.

## 5.2 SQFA for digit classification with poor first-order information

To examine how SQFA performs on high-dimensional real-world data, we tested it using the grayscale Street View House Numbers (SVHN) dataset, composed of 1024-dimensional images. The many sources of variation in the images (e.g. digits with mixed contrast polarity, variation in background intensity), make first-order differences between classes a poor signal for discriminating between the digits, and make second-order statistics important for classification performance. For each filter-learning method, we learned 2, 4, 8, and 16 filters ($m$). We then evaluated the QDA classification accuracy in each low-dimensional feature space (note that LDA cannot have $m > c - 1$, where the number of classes $c$ is 10 for this dataset).

Notably, all variants of SQFA achieved higher QDA accuracy than the remaining methods for all values of $m$ (Table 1). SQFA and SQFA-H were in general the best performing methods, being outperformed by SQFA-J only for $m = 16$. It is noteworthy that SQFA and SQFA-H both consistently outperform SQFA-B, which is the more commonly used dissimilarity measure in the dimensionality reduction literature (Choi & Lee, 2003; Duin & Loog, 2004). smSQFA performed similarly to SQFA, in line with the class-conditional means

Table 1: SVHN median QDA classification accuracy (%). Highest two accuracies are shown in bold.

| $m$ | SQFA | smSQFA | SQFA-H | SQFA-B | SQFA-W | SQFA-J | LDA | SPCA | LFDA | WDA | LMNN | PCA |
|---|---|---|---|---|---|---|---|---|---|---|---|---|
| 2 | **39.8** | 36.7 | **39.5** | 36.5 | 33.2 | 36.5 | 23.6 | 19.6 | 22.9 | 22.3 | 19.5 | 19.6 |
| 4 | **56.4** | **56.4** | **56.9** | 55.9 | 48.8 | 55.6 | 27.4 | 22.8 | 30.6 | 39.9 | 25.9 | 24.5 |
| 8 | **68.1** | 67.5 | **68.1** | 66.3 | 58.5 | 67.3 | 34.8 | 37.5 | 33.5 | 57.1 | 49.9 | 35.6 |
| 16 | 74.7 | 74.6 | **74.8** | 74.4 | 71.8 | **75.2** | – | 50.4 | 34.4 | 66.1 | 67.5 | 58.4 |

Table 2: MNIST median QDA classification accuracy (%). Highest two accuracies are shown in bold.

| $m$ | SQFA | smSQFA | SQFA-H | SQFA-B | SQFA-W | SQFA-J | LDA | SPCA | LFDA | WDA | LMNN | PCA |
|---|---|---|---|---|---|---|---|---|---|---|---|---|
| 2 | 59.7 | **62.4** | **66.6** | 56.0 | 52.6 | 50.1 | 56.6 | 48.4 | 59.2 | 55.0 | 52.6 | 46.1 |
| 4 | 80.0 | 76.6 | **86.6** | **82.5** | 74.6 | 76.8 | 82.5 | 72.9 | 82.2 | 67.8 | 73.5 | 63.1 |
| 8 | 89.7 | 87.8 | **93.2** | **90.8** | 88.7 | 88.7 | 90.1 | 88.5 | 89.4 | 86.8 | 90.2 | 86.5 |
| 16 | 94.2 | 94.0 | **95.4** | **94.4** | 94.1 | 94.0 | – | 92.5 | 90.5 | 93.8 | 94.2 | 93.7 |

containing little information for this dataset. Remarkably, this pattern of results was almost identical when evaluated with kNN accuracy, with SQFA and SQFA-H being the best performing methods (Appendix H).

### 5.3 SQFA for digit classification with useful first-order information

To examine how SQFA performs when both first- and second-order statistics are informative, we compared the same methods using the MNIST dataset. Because MNIST digits are white digits on a black background, the class-conditional means are quite different, making them useful for classification. The same number of filters were learned for each method as for the SVHN dataset.

SQFA-H achieved the highest QDA accuracy for all values of $m$ (Table 2). For most values of $m$, SQFA-B had the second highest performance. SQFA was competitive, performing better than many other methods (it placed either 3rd or 5th across values of $m$, out of 12 methods). Thus, when the class-conditional means are informative, maximizing the Fisher-Rao distance can be competitive, but maximizing the Hellinger distance can be the best option. The conclusions are the same when using kNN classifier accuracy (Appendix H).

### 5.4 Naturalistic speed-estimation task

Next, we examine how SQFA features perform on a naturalistic speed-estimation dataset used to investigate neural computations (Burge & Geisler, 2015; Chin & Burge, 2020; Herrera-Esposito & Burge, 2024). This video-based dataset is interesting for several reasons. First, finding features that are useful for solving visual tasks is essential for sensory-perceptual neuroscience (Burge, 2020), and a potential application for SQFA. Second, the class-conditional distributions are well approximated by zero-mean Gaussians, which means that the assumptions of smSQFA are approximately satisfied. Thus, the results with smSQFA for this dataset are a good approximation to the filters obtained by maximizing the true Fisher-Rao distance, as opposed to the Calvo-Oller bound used by SQFA. Third, a method called AMA-Gauss which directly maximizes the performance of a Bayesian Gaussian decoder is reported to perform well for this dataset (Jaini & Burge, 2017), providing a principled benchmark for comparison (see Appendix J).

Each video consists of 30 horizontal pixels and 15 frames. The vertical dimension was averaged out, hence the videos can be represented as 2D space-time plots (Figure 4A). Each video shows a naturally textured surface moving with one of 41 different speeds (i.e. classes). We learned up to 8 filters with each method, following the original work (Burge & Geisler, 2015).

As expected, AMA-Gauss performed best, since it directly optimizes the decodability of speed under Gaussian assumptions (Table 3). smSQFA is the second-best method for most values of $m$, and SQFA-H was also a top-ranking method across values of $m$. SQFA-B achieved high performance, but not as good as smSQFA

Table 3: Speed estimation median QDA classification accuracy (%). Highest two accuracies are shown in bold.

| $m$ | SQFA | smSQFA | SQFA-H | SQFA-B | SQFA-W | SQFA-J | AMA | LDA | SPCA | LFDA | WDA | LMNN | PCA |
|---|---|---|---|---|---|---|---|---|---|---|---|---|---|
| 2 | 58.1 | **58.8** | 56.0 | 58.1 | 31.3 | 58.7 | **61.9** | 5.4 | 9.5 | 4.9 | 9.4 | 4.7 | 23.9 |
| 4 | 68.0 | **68.3** | 67.3 | 66.4 | 45.4 | 65.8 | **74.1** | 11.2 | 31.3 | 25.3 | 18.0 | 15.3 | 32.8 |
| 6 | 80.7 | 79.6 | **83.9** | 78.3 | 61.5 | 68.5 | **85.9** | 20.9 | 40.3 | 11.5 | 30.3 | 35.3 | 48.7 |
| 8 | 89.3 | **89.4** | 89.2 | 84.9 | 80.8 | 69.9 | **91.9** | 28.1 | 59.8 | 52.8 | 38.6 | 51.0 | 75.1 |

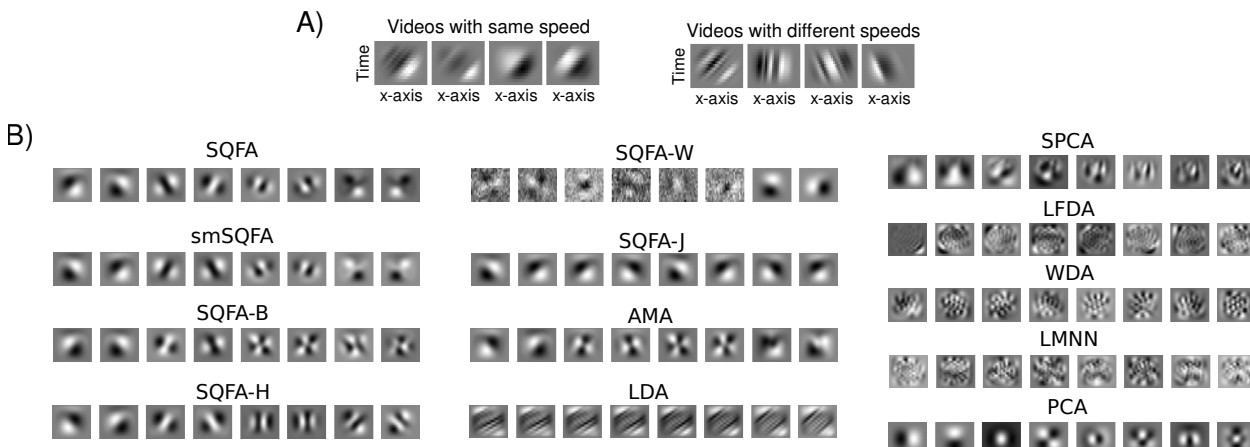

Figure 4: SQFA learns motion-sensitive features similar to those found in biological neural systems. **A)** 4 example videos with the same speed (left) and 4 example videos with different speeds (right). Each video is shown as a 2D space-time plot where the vertical axis is time and the horizontal axis is space. **C)** Filters learned by the methods (each image shows a 2D space-time plot).

and SQFA-H. Similar conclusions are obtained when using the kNN classifier instead of QDA (Appendix H, although see the accompanying discussion). We note that SQFA variants do not achieve as high QDA accuracy as AMA-Gauss filters because of two reasons: the dissimilarity measures used by SQFA variants are not perfect measures of decodability, and the average of pairwise discriminabilities is not guaranteed to maximize multiclass discriminability (Loog et al., 2001; Thangavelu & Raich, 2008) (see Section D.1). The fact that SQFA and SQFA-H perform so close to AMA-Gauss highlights the utility of Fisher-Rao and Hellinger distances as dimensionality reduction objectives.

Finally, we visualized the features learned by the different methods. To obtain more interpretable filters, we used pairwise filter learning when visualizing the SQFA variants (see Section 4). The best performing SQFA variants (SQFA, smSQFA, SQFA-H and SQFA-B), as well as AMA-Gauss, learned filters that are similar to typical motion-sensitive receptive fields in visual cortex, selecting for a range of spatio-temporal frequencies (Movshon et al., 1978; Rust et al., 2005; Priebe et al., 2006). In contrast, filters learned by all other methods either lack clear motion sensitivity, or do not cover a range of spatio-temporal frequencies (Figure 4C). This shows that SQFA can learn features that are both useful for classification and interpretable.

## 5.5 Neural data analysis

Next, to show the applicability of SQFA to real-world experimental data, we tested SQFA on an open dataset of neural spike recordings (Zandvakili & Kohn, 2015; Amin Zandvakili & Adam Kohn, 2019). SQFA can be useful to analyze neural data for different reasons. First, because stimulus-dependent (i.e. class-specific) response covariances are considered important for neural coding (Moreno-Bote et al., 2014; Kohn et al., 2016), and finding the informative low-dimensional subspaces that account for stimulus-dependent covariances is an important problem in neuroscience. Second, there is interest in studying the geometry of neural representations (Kriegeskorte & Wei, 2021; Wang & Ponce, 2021; Chung & Abbott, 2021), for which information geometry can provide useful tools (Kriegeskorte & Wei, 2021). It is also worth noting that spiking neural responses present an interesting challenge for SQFA, because they are noisy and highly non-Gaussian.

The responses were recorded from primary visual cortex of macaque monkeys shown drifting gratings of 8 different orientations (classes). The dataset includes 5 recording sessions from 3 animals, each session with 400 trials per class, and between 70 and 142 recorded neurons. We removed outlier trials (1.2%) and neurons (3.3%) (see Appendix K). Because this dataset consists of different sessions that cannot be used to learn a single set of features, we used a modified evaluation procedure. For each session we performed 5 different

Table 4: Neural data mean QDA classification accuracy (%). Highest two accuracies are shown in bold.

| $m$ | SQFA | smSQFA | SQFA-H | SQFA-B | SQFA-W | SQFA-J | LDA | SPCA | LFDA | WDA | LMNN | PCA |
|---|---|---|---|---|---|---|---|---|---|---|---|---|
| 2 | 90.2 | 88.1 | **91.4** | 88.4 | 86.0 | 86.9 | 88.7 | 85.6 | 88.1 | 88.0 | **91.0** | 68.4 |
| 4 | 94.4 | 94.2 | **96.1** | 93.2 | 85.4 | 92.1 | 93.8 | 92.5 | 93.6 | 91.7 | **95.0** | 91.5 |
| 8 | 96.0 | 95.4 | **96.8** | **96.5** | 83.1 | 95.7 | – | 96.4 | 95.9 | 87.2 | **96.7** | 96.4 |

random stratified splits of the data into training, validation, and test sets, resulting in 25 total splits. For each split we learned the filters and evaluated QDA classification accuracy. We report the mean QDA accuracy across all splits[6]. For more information about the dataset see Amin Zandvakili & Adam Kohn (2019).

Similar to previous datasets, SQFA-H had the highest performance for all values of $m$ (Table 4). LMNN had the second highest performance, and SQFA was the third-best performing method for $m = 2$ and $m = 4$, again proving that maximizing the Fisher-Rao distances is a competitive strategy for dimensionality reduction. Similar conclusions are obtained when using the kNN classifier (Appendix H). These results show that SQFA and its variants can be useful for finding low-dimensional subspaces of neural data that make the best use of stimulus-dependent covariances for classification. Additionally, the results show that SQFA can perform well even when the data are highly non-Gaussian.

### 5.6 Maximizing the Calvo-Oller bound leads to largest Fisher-Rao distances

Lastly, we evaluated the performance of the Calvo-Oller bound as a surrogate for the Fisher-Rao distance in the learning objective. First, we asked whether maximizing the Calvo-Oller bound leads to the largest Fisher-Rao distance among the different objectives. For the filters learned in each of the datasets above by LDA, PCA, and each SQFA variant, we numerically computed the average Fisher-Rao distance between the class distributions in the learned feature spaces using the method of Nielsen (2023), implemented in the package `pyBregMan` (Nielsen & Soen, 2024). For all datasets and values of $m$, maximizing the Calvo-Oller bound led to the highest average Fisher-Rao distance in the feature space (Table 5).

Table 5: Mean Fisher–Rao class distance in feature space for different optimization objectives. Columns indicate the distance maximized by the SQFA variants, or the learning algorithm.

| $m$ | Dataset | Calvo–Oller | Hellinger | Bhattacharyya | Wasserstein | Jeffreys | PCA | LDA |
|---|---|---|---|---|---|---|---|---|
| 2 | SVHN | **1.42** | 1.20 | 1.35 | 1.20 | 1.36 | 0.12 | 0.37 |
| 4 | SVHN | **2.31** | 2.24 | 2.29 | 1.87 | 2.27 | 0.55 | 0.92 |
| 8 | SVHN | **2.51** | **2.51** | 2.45 | 2.12 | 2.47 | 1.13 | 1.07 |
| 16 | SVHN | **3.50** | 3.49 | 3.49 | 3.05 | 3.44 | 2.26 | – |
| 2 | MNIST | **2.92** | 2.86 | 2.80 | 2.66 | 2.73 | 2.40 | 2.22 |
| 4 | MNIST | **3.90** | 3.79 | 3.84 | 3.54 | 2.77 | 3.21 | 3.16 |
| 8 | MNIST | **3.85** | 3.49 | 3.56 | 3.66 | 3.76 | 3.56 | 3.02 |
| 16 | MNIST | **4.62** | 4.31 | 4.55 | 4.40 | 4.42 | 4.34 | – |
| 2 | Speed | **3.34** | **3.34** | **3.34** | 2.97 | 3.34 | 2.81 | 1.30 |
| 4 | Speed | **4.33** | 4.27 | 4.32 | 3.97 | 3.91 | 3.72 | 3.09 |
| 8 | Speed | **5.36** | 5.19 | 5.27 | 5.09 | 4.51 | 4.86 | 5.26 |
| 2 | Neural data | **3.81** | 3.78 | 3.75 | 3.47 | 3.78 | 2.95 | 3.79 |
| 4 | Neural data | **4.83** | 4.56 | 4.68 | 4.00 | 4.81 | 4.55 | 4.56 |
| 8 | Neural data | **5.42** | 5.18 | 4.91 | 5.30 | 5.40 | 4.98 | – |

Then, we compared the Calvo-Oller bounds to the Fisher-Rao distances in the SQFA feature space across all class pairs. Figure 5 shows the comparison for $m = 8$. The pairwise Calvo-Oller bounds are highly correlated with the true Fisher-Rao distances in all datasets. Furthermore, the Calvo-Oller bounds are close to the true

---

[6]We use the mean for this dataset because performance across splits is highly correlated within sessions. The mean better summarizes the performance across sessions than the median, which reflects only the session with the middle performance. The same conclusion are obtained when using the median.

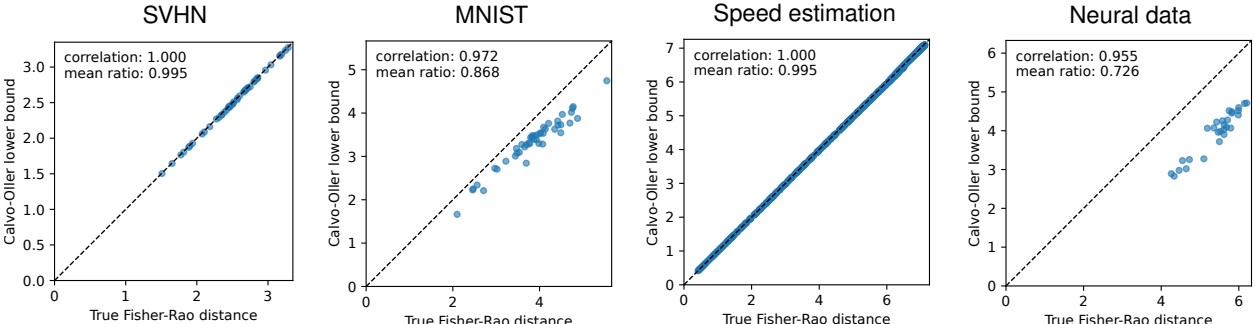

Figure 5: Calvo-Oller bound vs. true Fisher-Rao distance in real-world datasets. Each panel shows the Calvo-Oller bound and the Fisher-Rao distance for the class pairs of a given dataset. From left to right, the SVHN, MNIST and the speed estimation dataset are shown. Each point represents a different pair of classes. The dashed line shows the identity line. The correlation and the mean of the pairwise ratios between the Calvo-Oller bound and the true Fisher-Rao distance are shown in the top left of each panel.

Fisher-Rao distance, as indicated by the mean ratios between the two. The largest deviation was observed for the neural dataset, which has a mean ratio of 0.726, indicating that the Calvo-Oller bound was on average 27.4% smaller than the true Fisher-Rao distance. The Calvo-Oller bound and the Fisher-Rao distance are almost identical for the SVHN and speed estimation datasets, which is expected because these datasets have similar means across classes, and the Calvo-Oller bound is exact in the equal-mean case (Appendix B).

In sum, these results show that maximizing the Calvo-Oller bound is a practical way to maximize the Fisher-Rao distance for learning in real-world datasets.

## 6 Discussion

We have introduced SQFA, a supervised dimensionality reduction method that maximizes the Fisher-Rao distances between class-conditional distributions under Gaussian assumptions. SQFA is a computationally efficient method that learns features supporting excellent quadratic decodability, i.e. QDA classification accuracy. The same optimization procedure can be used to define variants of SQFA that maximize other dissimilarity measures from statistics and information theory.

The results exhibit noteworthy patterns. First, maximizing the Fisher-Rao distance, which is derived in a geometric framework, leads to features that support similar performance as maximizing the Bhattacharyya and Hellinger distances, which are information-theoretic dissimilarity measures directly linked to Bayes classification error. Also, SQFA features often supported higher QDA and kNN accuracy than popular dimensionality reduction methods such as LDA, SPCA, LFDA, WDA and LMNN. Therefore, SQFA is a competitive method for supervised dimensionality reduction that can be useful in combination with different types of decoders. Notably, maximizing the Fisher-Rao distance led to much better discriminative features than maximizing the Wasserstein distance (SQFA-W). Hence, Riemannian distances between class-conditional distributions that do not reflect discriminability (e.g. Wasserstein) are not necessarily good objectives for learning discriminative features.

Second, despite the fact that the Hellinger and Bhattacharyya distances are equivalent objectives for the two-class case, SQFA-H consistently outperformed SQFA-B in our analyses. Moreover, SQFA-H was in general the best performing method across datasets and number of filters learned, both for QDA and for kNN accuracy. Why does maximizing the Hellinger distance lead to better performance than maximizing the Bhattacharyya distance? One likely explanation relates to the fact that the sum of pairwise discriminabilities does not perfectly translate to multiclass discriminability (Loog et al., 2001; Tao et al., 2007; Thangavelu & Raich, 2008). In particular, the behavior of the multiclass objective depends on how it weights the discriminabilities of the different pairs of classes. Because dissimilarity measures can scale differently as

distributions become further apart, the relative contributions of the class pairs to the objective in Equation 1 can be different (see Section 3.1). For example, the Hellinger distance is bounded between 0 and 1, and classes that are already well separated will not contribute much to the loss gradient, so the learning procedure will emphasize separating classes that are close together. In contrast, the Bhattacharyya distance grows faster as two classes are further apart, which can favor separating classes that are already well separated, leading to little improvement in classification performance (see Appendix D.1). The advantage of the Hellinger distance over the Bhattacharyya distance observed in our analyses should be of interest for practitioners, because the Bhattacharyya distance is the most common choice in linear dimensionality reduction. Notably, the Fisher-Rao distance often grows at a rate between those of the Bhattacharyya and the Hellinger distances, which might help explain the good performance of SQFA.

A novel methodological aspect of this work is the use of the Calvo-Oller bound as a surrogate for the Fisher-Rao distance in learning (Calvo & Oller, 1990; Nielsen, 2023). The fact that closed-form expressions for the Fisher-Rao distance between multivariate Gaussians are unavailable has limited its use in practice. Besides the success of the Calvo-Oller bound at learning discriminative features, we showed that maximizing the Calvo-Oller bound led to a higher Fisher-Rao distance than maximizing any other objective, and that the pairwise Calvo-Oller bounds were highly correlated with the true pairwise Fisher-Rao distances. This motivates further explorations of the Calvo-Oller bound as an objective for machine learning applications. Additionally, formulas for the Calvo-Oller bound exist for other elliptical distributions such as multivariate Student-t and Cauchy distributions (Calvo & Oller, 2002; Nielsen, 2023). Future work can use these formulas to extend SQFA to other non-Gaussian elliptical distributions.

Information geometry is a promising tool for studying neural representations in neuroscience and machine learning (Kriegeskorte & Wei, 2021; Wang & Ponce, 2021; Arvanitidis et al., 2022; Duong et al., 2023; Feather et al., 2024). Finding features that maximize Fisher-Rao distances between classes, or conditions, is a potentially useful tool in this context. Recent research in neuroscience and psychology suggests a relation between geodesic distances in perceptual space (using a metric analogous to the Fisher metric) and similarity judgments (Riemann, 1867; Fechner, 1860; Bujack et al., 2022; Zhou et al., 2024; Vacher & Mamassian, 2024; Hong et al., 2025). SQFA shows that maximizing a geometric objective with similar distances can be useful for supervised dimensionality reduction.

SQFA is a first step towards using information geometry for dimensionality reduction (also see Carter et al. (2009; 2011); Dwivedi et al. (2022)). Several questions and extensions remain open. For example, promising directions include extending SQFA to non-Gaussian elliptical distributions, or even to the non-parametric case, using the non-parametric Fisher-Rao distance (Srivastava et al., 2007). Future work should also study under what circumstances maximizing the Fisher-Rao distance might be preferable to maximizing other measures of dissimilarity, such as the Hellinger distance. Finally, it should be straightforward to extend this framework to nonlinear feature learning, a topic that will be explored in future work.

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

# A    Fisher-Rao distance and discriminability in the zero-mean case

## A.1    Generalized eigenvalues reflect quadratic differences between classes

As described in the main text, the Fisher-Rao distance between two zero-mean Gaussian distributions $\mathcal{N}(0, \boldsymbol{\Psi}_i)$ and $\mathcal{N}(0, \boldsymbol{\Psi}_j)$ is given (up to a factor of $\sqrt{2}$) by the affine-invariant distance in the manifold of symmetric positive definite matrices, SPD(m)

$$d_{FR}(\boldsymbol{\Psi}_i, \boldsymbol{\Psi}_j) = \sqrt{\frac{1}{2} \sum_{k=1}^{m} \log^2(\lambda_k)} \tag{7}$$

where $\lambda_k$ are the generalized eigenvalues of the pair of matrices $(\boldsymbol{\Psi}_i, \boldsymbol{\Psi}_j)$.

The generalized eigenvalues $\lambda_k$ and generalized eigenvectors $\mathbf{v}_k$ of the pair of matrices $\mathbf{A} \in \text{SPD}(m)$ and $\mathbf{B} \in \text{SPD}(m)$ are the solutions to the generalized eigenvalue problem $\mathbf{A}\mathbf{v}_k = \lambda_k \mathbf{B}\mathbf{v}_k$. The solution to the problem is given by the eigenvalues and eigenvectors of $\mathbf{B}^{-1/2}\mathbf{A}\mathbf{B}^{-1/2}$, where $\mathbf{B}^{-1/2}$ is the inverse square root of $\mathbf{B}$. If $\mathbf{A}$ and $\mathbf{B}$ are identical, $\mathbf{B}^{-1/2}\mathbf{A}\mathbf{B}^{-1/2}$ is the identity matrix, all the eigenvalues are 1, and $d_{FR}(\mathbf{A}, \mathbf{B}) = 0$. The farther the $\lambda_k$ are from 1, the more different the matrices $\mathbf{A}$ and $\mathbf{B}$ are (i.e. the more different $\mathbf{B}^{-1/2}\mathbf{A}\mathbf{B}^{-1/2}$ is from the identity matrix).

Consider a random variable $\mathbf{z} \in \mathbb{R}^m$ that belongs to one of two classes $i, j$, with second moment matrices $\boldsymbol{\Psi}_i = \mathbb{E}\left[\mathbf{z}\mathbf{z}^T | y = i\right]$ and $\boldsymbol{\Psi}_j = \mathbb{E}\left[\mathbf{z}\mathbf{z}^T | y = j\right]$. Next, consider a vector $\mathbf{w} \in \mathbb{R}^m$ and the squared projection of $\mathbf{z}$ onto $\mathbf{w}$, $(\mathbf{w}^T\mathbf{z})^2$. The following ratio relates to how different the squared projections are for the two classes, which is a useful proxy for quadratic discriminability:

$$R(\mathbf{w}) = \frac{\mathbb{E}[(\mathbf{w}^T\mathbf{z})^2 | y = i]}{\mathbb{E}[(\mathbf{w}^T\mathbf{z})^2 | y = j]} = \frac{\mathbf{w}^T \boldsymbol{\Psi}_i \mathbf{w}}{\mathbf{w}^T \boldsymbol{\Psi}_j \mathbf{w}} \tag{8}$$

The local extrema of the ratio $R(\mathbf{w})$ are obtained at the generalized eigenvectors $\mathbf{w} = \mathbf{v}_k$ of $(\boldsymbol{\Psi}_i, \boldsymbol{\Psi}_j)$, where the ratio $R(\mathbf{v}_k) = \lambda_k$ (Fukunaga, 1990).

The more different $\lambda_k$ is from 1, the more different are the expected squared projections $(\mathbf{v}^T\mathbf{z})^2$ for the two classes. The magnitude of $\log^2 \lambda_k$ indicates how different the ratio in Equation 8 is from 1, in proportional terms. The set of $\mathbf{v}_k$'s spans the space of $\mathbf{z}$, so $d_{FR}(\boldsymbol{\Psi}_i, \boldsymbol{\Psi}_j)$ summarizes the quadratic differences between the classes $i, j$ along all directions in the feature space, thus relating to the quadratic discriminability of the classes.

Of course, the quadratic discriminability between the classes depends on factors other than the ratio of the expected values of the squared projections, and larger differences in this ratio do not strictly indicate higher discriminability. However, empirical studies show that the generalized eigenvalues tend to be a good indicator of quadratic discriminability in real world datasets (Karampatziakis & Mineiro, 2014).

## A.2    Fisher-Rao distance and Bayes error

Here, we explicitly compare the Fisher-Rao distance to the Bayes error for zero-mean Gaussians in the 1D and 2D cases.

Two symmetric positive definite (SPD) matrices $\boldsymbol{\Psi}_i$ and $\boldsymbol{\Psi}_j$ can be simultaneously diagonalized by a linear transformation of the data space. Given the affine-invariance of the Fisher-Rao distance and of the Bayes error, we can reduce their analysis to the analysis of Gaussians of the form $\mathcal{N}(0, \mathbf{D})$ and $\mathcal{N}(0, \mathbf{I})$, where $\mathbf{D}$ is a diagonal matrix and $\mathbf{I}$ is the identity. Then, the Fisher-Rao and the Bayes error are a function of the diagonal elements of $\mathbf{D}$. The diagonal elements of $\mathbf{D}$ are the generalized eigenvalues of $(\mathbf{D}, \mathbf{I})$, so the Fisher-Rao distance between $\mathcal{N}(0, \mathbf{D})$ and $\mathcal{N}(0, \mathbf{I})$ is given by

$$d_{FR}(\mathbf{D}, \mathbf{I}) = \sqrt{\frac{1}{2} \sum_{k=1}^{m} \log^2(\mathbf{D}_{kk})} \tag{9}$$

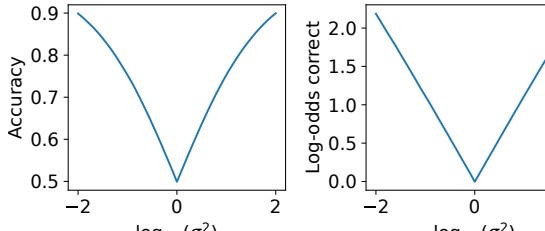 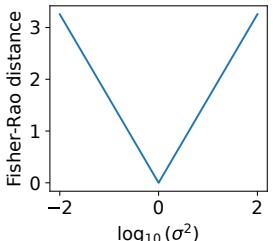

Figure 6: Distances and Bayes error for 1D Gaussian distributions. From left to right, the three panels show, as a function of $\log_{10}(\sigma^2)$, accuracy of the Bayes classifier, the log-odds ratio of correct classification, and the Fisher-Rao distance.

We computed the Bayes error ($e_B$) for each value of $\mathbf{D}$ by simulating 100,000 samples from each $\mathcal{N}(0, \mathbf{D})$ and $\mathcal{N}(0, \mathbf{I})$ distribution, and obtaining the error rate of the optimal probabilistic classifier. We compute the Bayes accuracy ($a_B$) as $a_B = 1 - e_B$. Because the accuracy is bounded to be between 0 and 1, we also computed the log-odds of correctly classifying a sample, given by $\log\left(\frac{a_B}{e_B}\right)$.

**Zero-mean Gaussian, 1D case.** In the 1D case, the matrix $\mathbf{D}$ is a single positive number $\sigma^2$. Figure 6 shows the Bayes error, the log-odds of correct classification, and the Fisher-Rao distance as a function of $\log \sigma^2$.

The Fisher-Rao distance grows linearly with $|\log_{10} \sigma^2|$ in the 1D case. Interestingly, the Bayes accuracy is approximately linear for small values of $\sigma^2$, although it begins to saturate for larger values of $\sigma^2$ (since it is bounded by 1). The log-odds of a correct classification also looks like a linear function of $|\log_{10} \sigma^2|$. This suggests that the Fisher-Rao distance is a good proxy for classification performance in the zero-mean 1D Gaussian case.

**Zero-mean Gaussian, 2D case.** Next we analyze the 2D case. Here, there are two parameters, $\sigma_1^2$ and $\sigma_2^2$. The top row of Figure 7 shows the level sets of accuracy, the log-odds correct, and the Fisher-Rao distance as a function of $\log_{10} \sigma_1^2$ and $\log_{10} \sigma_2^2$. The level sets of the Fisher-Rao distance have a different shape than the level sets of the accuracy and log-odds correct. The contours of the accuracy and log-odds correct have more circular shapes close to the origin, but acquire a tilted hexagonal shape farther from the origin. As expected from Equation 7, the Fisher-Rao contour sets are circles.

These contour shapes show that the Fisher-Rao distance is not a perfect proxy for discriminability, since it does not capture the interactions between the two parameters $\log_{10} \sigma_1^2$ and $\log_{10} \sigma_2^2$ that lead to the distinctive hexagonal shape of the accuracy and log-odds correct plots. The Fisher-Rao distance provides a good approximation of the log-odds correct, however, at smaller values of $\log_{10} \sigma_1^2$ and $\log_{10} \sigma_2^2$.

To visualize this further, we plotted the values of the distances and accuracies for three different slices of the 2D space, with fixed values of $\log_{10} \sigma_2^2$ (0.0, 0.56, 1.12). (Figure 7, middle row). For the first two slices (black and red), the Fisher-Rao distance and the log-odds correct are very similar across the range of values of $\log_{10} \sigma_1^2$, but they become less similar for the third slice (blue), where the asymmetry of the log-odds correct is more pronounced.

In sum, the Fisher-Rao distance broadly captures the dependence of discriminability on the distribution parameters in some scenarios, particularly in the 1D case, and in the 2D case when the generalized eigenvalues are closer to 1 (i.e. when $\log_{10} \sigma^2$ is close to 0), but that it also fails to capture some patterns of the dependence of discriminability on the parameters.

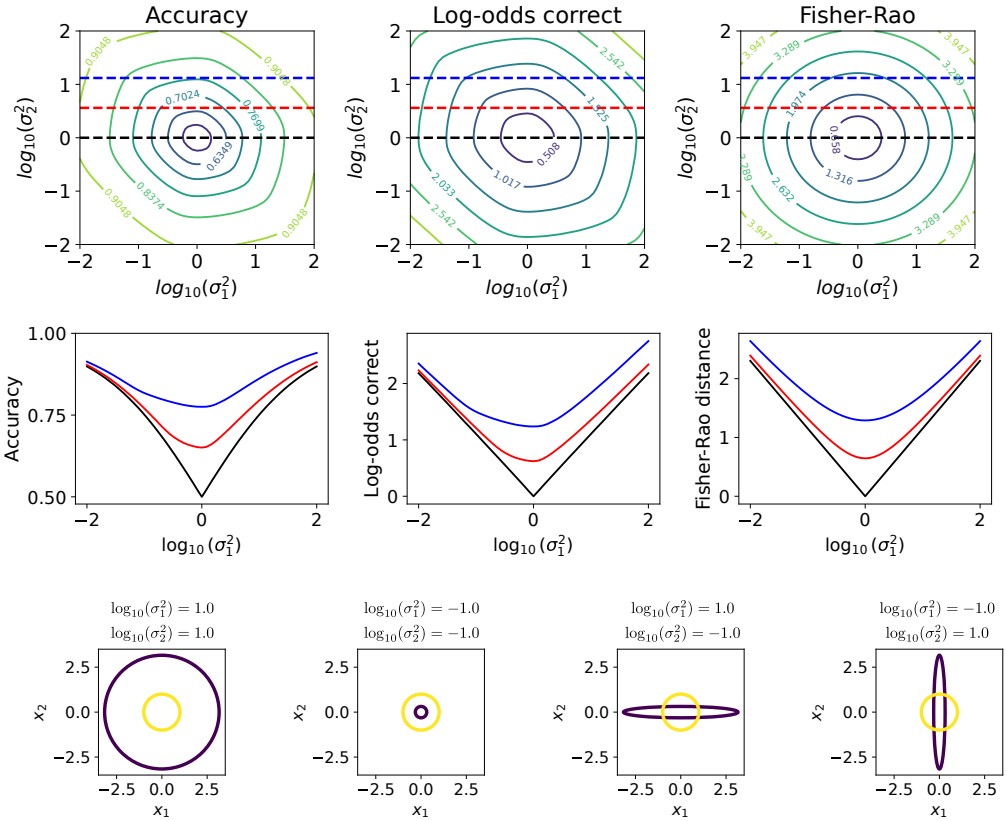

Figure 7: Distances and Bayes error for 2D Gaussian distributions. From left to right, the panels show for two Gaussian classes, the accuracy of the Bayes classifier, the log-odds of a correct classification, and the Fisher-Rao distance. **Top**. Contour plots of the quantities as a function of $\log_{10}(\sigma_1^2)$ and $\log_{10}(\sigma_2^2)$. The values for the contour lines are shown as a color map. The horizontal dashed lines indicate the 1D slices that are shown in the middle row. **Middle**. The same quantities as the top row, shown as a function of $\log_{10}(\sigma_1^2)$, for fixed values of $\log_{10}(\sigma_2^2)$. The values used are 0.0 (black), 0.56 (red), and 1.12 (blue). **Bottom**. Examples of two zero-mean distributions for different values of $\log_{10}\sigma_1^2$ and $\log_{10}\sigma_2^2$.

# B  Calvo-Oller bound as a surrogate for the Fisher-Rao distance

In this section we discuss the Calvo-Oller bound in special cases where we have exact formulas for the Fisher-Rao distance (see Nielsen (2023)).

**Equal mean case.**  For Gaussians with equal mean, the Calvo-Oller bound is equal to the true Fisher-Rao distance. First, we can use the affine-invariant property of the Calvo-Oller bound (Nielsen, 2023) to subtract the common mean from the data, thus reducing the problem to the zero-mean case. Then, given that Calvo-Oller embedding for class $i$ is $\mathbf{\Omega}_i = \begin{bmatrix} \mathbf{\Sigma}_i & \mathbf{0} \\ \mathbf{0}^T & 1 \end{bmatrix}$, it is easy to see that $\mathbf{\Omega}_i^{-1}\mathbf{\Omega}_j = \begin{bmatrix} \mathbf{\Sigma}_i^{-1}\mathbf{\Sigma}_j & \mathbf{0} \\ \mathbf{0}^T & 1 \end{bmatrix}$, and that the generalized eigenvalues of $(\mathbf{\Omega}_i, \mathbf{\Omega}_j)$ are the eigenvalues of $\mathbf{\Sigma}_i^{-1}\mathbf{\Sigma}_j$ plus an additional generalized eigenvalue equal to 1. Since both the Calvo-Oller distance and the Fisher-Rao distance are given by the sum of the squared logarithm of the generalized eigenvalues (Equation 4), it follows that the Calvo-Oller distance is equal to the true Fisher-Rao distance.

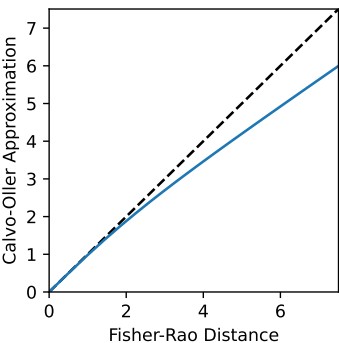

Figure 8: Calvo-Oller bound vs. true Fisher-Rao distance for the equal-covariance case. The Fisher-Rao distance and the Calvo-Oller distance were computed for a range of Mahalanobis distances ranging from 0 to 20. The dashed line shows the identity line.

**Equal covariance case.** For two Gaussians with equal covariance, $\mathcal{N}(\boldsymbol{\mu}_i, \boldsymbol{\Sigma})$ and $\mathcal{N}(\boldsymbol{\mu}_j, \boldsymbol{\Sigma})$, we denote the squared Mahalanobis distance as

$$d_M(\boldsymbol{\mu}_i, \boldsymbol{\mu}_j)^2 = (\boldsymbol{\mu}_i - \boldsymbol{\mu}_j)^T \boldsymbol{\Sigma}^{-1} (\boldsymbol{\mu}_i - \boldsymbol{\mu}_j) \tag{10}$$

Then, the exact Fisher-Rao distance is given by (Nielsen, 2023)

$$d_{FR}(\boldsymbol{\mu}_i, \boldsymbol{\mu}_j) = \sqrt{2}\text{arccosh}\left(1 + \frac{1}{4}d_M(\boldsymbol{\mu}_i, \boldsymbol{\mu}_j)^2\right) \tag{11}$$

Figure 8 shows the Calvo-Oller bound as a function of the true Fisher-Rao distance for Gaussians with identity covariance, ranging from Mahalanobis distance of 0 to 20. Across this large range of Mahalanobis distances, the Calvo-Oller bound is close to the true Fisher-Rao distance.

## C  LDA features maximize squared Mahalanobis distances

Here, we prove that Linear Discriminant Analysis (LDA) maximizes the squared Mahalanobis distances between classes when the classes are homoscedastic Gaussians. The Mahalanobis distance equals the Fisher-Rao distance along the submanifold of Gaussians with equal covariance[7], which means that LDA maximizes the pairwise Fisher-Rao (squared) distances between classes along that submanifold.

For a labeled random variable $\mathbf{x} \in \mathbb{R}^n$ with class labels $y \in \{1, \ldots, c\}$, the goal of LDA is to find the filters $\mathbf{F} \in \mathbb{R}^{n \times m}$ such that the variable $\mathbf{z} = \mathbf{F}^T \mathbf{x}$ maximizes the between-class scatter relative to the within-class scatter. This is typically formulated as maximizing the Fisher criterion

$$\text{Tr}\left(\boldsymbol{\Sigma}^{-1} \mathbf{S_z}\right) \tag{12}$$

where $\boldsymbol{\Sigma}$ is the residual within-class covariance matrix of the data (i.e. the covariance matrix of the data after subtracting the class mean from each data point), and $\mathbf{S_z}$ is the between-class scatter matrix of $\mathbf{z}$, defined as

$$\mathbf{S_z} = \frac{1}{c}\sum_{i=1}^{c}(\boldsymbol{\mu}_i - \boldsymbol{\mu})(\boldsymbol{\mu}_i - \boldsymbol{\mu})^T$$

where $\boldsymbol{\mu}_i$ is the mean of class $i$ in the feature space and $\boldsymbol{\mu} = \sum_{i=1}^{c} \boldsymbol{\mu}_i$.

---

[7]This is not the same as the Fisher-Rao distance along the general manifold of Gaussian distributions.

The sum of pairwise squared Mahalanobis distances between the classes is given by

$$\frac{1}{2}\sum_{i=1}^{c}\sum_{j=1}^{c}d_M(\boldsymbol{\mu}_i,\boldsymbol{\mu}_j)^2 = \frac{1}{2}\sum_{i=1}^{c}\sum_{j=1}^{c}(\boldsymbol{\mu}_i-\boldsymbol{\mu}_j)^T\boldsymbol{\Sigma}^{-1}(\boldsymbol{\mu}_i-\boldsymbol{\mu}_j) \tag{13}$$

where the factor of $1/2$ is included to control for double-counting.

We can show that the objectives in Equation 12 and Equation 13 are equivalent. First, we assume without loss of generality that the overall mean $\boldsymbol{\mu} = \sum_{i=1}^{c}\boldsymbol{\mu}_i = 0$. Then, we have

$$\begin{aligned}
\frac{1}{2}\sum_{i=1}^{c}\sum_{j=1}^{c}d_M(\boldsymbol{\mu}_i,\boldsymbol{\mu}_j)^2 &= \frac{1}{2}\sum_{i=1}^{c}\sum_{j=1}^{c}(\boldsymbol{\mu}_i-\boldsymbol{\mu}_j)^T\boldsymbol{\Sigma}^{-1}(\boldsymbol{\mu}_i-\boldsymbol{\mu}_j) \\
&= \frac{1}{2}\sum_{i=1}^{c}\sum_{j=1}^{c}\left(\boldsymbol{\mu}_i^T\boldsymbol{\Sigma}^{-1}\boldsymbol{\mu}_i + \boldsymbol{\mu}_j^T\boldsymbol{\Sigma}^{-1}\boldsymbol{\mu}_j - 2\boldsymbol{\mu}_i^T\boldsymbol{\Sigma}^{-1}\boldsymbol{\mu}_j\right) \\
&= \sum_{i=1}^{c}\boldsymbol{\mu}_i^T\boldsymbol{\Sigma}^{-1}\boldsymbol{\mu}_i - \sum_{i=1}^{c}\sum_{j=1}^{c}\boldsymbol{\mu}_i^T\boldsymbol{\Sigma}^{-1}\boldsymbol{\mu}_j \\
&= \sum_{i=1}^{c}\boldsymbol{\mu}_i^T\boldsymbol{\Sigma}^{-1}\boldsymbol{\mu}_i - \left(\sum_{i=1}^{c}\boldsymbol{\mu}_i^T\right)\boldsymbol{\Sigma}^{-1}\left(\sum_{j=1}^{c}\boldsymbol{\mu}_j\right) \\
&= \sum_{i=1}^{c}\boldsymbol{\mu}_i^T\boldsymbol{\Sigma}^{-1}\boldsymbol{\mu}_i \\
&= \mathrm{Tr}\left(\boldsymbol{\Sigma}^{-1}\sum_{i=1}^{c}\boldsymbol{\mu}_i\boldsymbol{\mu}_i^T\right) \\
&= c\,\mathrm{Tr}\left(\boldsymbol{\Sigma}^{-1}\mathbf{S_z}\right)
\end{aligned}$$

In the first five lines we expanded the squared Mahalanobis distance, used the linearity of the dot product and the fact that $\boldsymbol{\mu} = 0$. In the last two lines we used the linearity and the cyclic property of the trace, and that $\sum_{i=1}^{c}\boldsymbol{\mu}_i\boldsymbol{\mu}_i^T = c\mathbf{S_z}$ because $\boldsymbol{\mu} = 0$. This shows that maximizing the LDA criterion maximizes the pairwise squared Mahalanobis distances between classes.

## D    Comparison of different distances

In this section, we compare the behavior of the Fisher-Rao distance with the two other main dissimilarity measures in this work: the Hellinger and the Bhattacharyya distance[8].

### D.1    Distances scale differently in the equal-covariance case

For the case of Gaussians with equal covariance and different means, all three distances have closed form expressions as a function of the Mahalanobis distance (Equation 11 and Equation 6).

Figure 2 in the main text shows that the three distances grow monotonically with the Mahalanobis distance, but with different growth rates. The Bhattacharyya distance equals the squared Mahalanobis distance in the equal covariance case (see Equation 6). As two classes become more separated, the Bhattacharyya distance grows faster. On the other hand, the Fisher-Rao distance grows sublinearly with the Mahalanobis distance. As two classes become more separated, the Fisher-Rao distance grows more slowly. Finally, the Hellinger distance is bounded unlike the two other distances. As two classes become more separated, the Hellinger distance approaches the maximum value of 1.

The difference in the growth rates has implications for their behavior as learning objectives. In the multi-class dimensionality reduction problem, the Bhattacharyya distance will favor separating classes that are already

---

[8]We remind the reader that the Bhattacharyya distance is not actually a distance in the geometric sense. We still refer to it as a distance, in line with the literature.

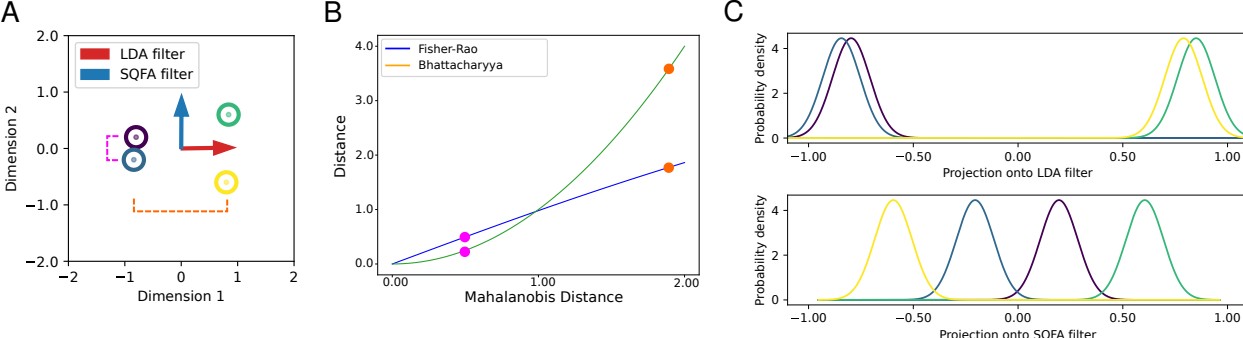

Figure 9: **A)** Toy dataset with 4 classes of 2D Gaussian distributions with identical covariance and different means. Arrows indicate the filters learned by LDA (red) and by SQFA (blue). **B)** Value of the Fisher-Rao distance (blue curve) and Bhattacharyya distance (green curve) for the distance between clusters along the horizontal axis (orange dot) and the spacing along classes along the vertical axis (magenta dot). **C** Projection of the classes onto the LDA (top) and SQFA (bottom) filters.

well-separated, since those lead to a larger increase in the average pairwise Bhattacharyya distance. On the other hand, the Hellinger distance will strongly favor separating classes that are close together, since further separating classes that are already far apart will barely change the average Hellinger distance. The Fisher-Rao distance will behave somewhere in between: it favors separating classes that are close together, but it is still sensitive to classes that are farther apart, since it is an unbounded distance.

We designed a toy dataset to illustrate how the different relative weights of the pairs of classes for the different objectives can affect the learning outcome. We learned one filter with SQFA and with LDA for this dataset. Because LDA maximizes the average squared Mahalanobis distance (Appendix C), it is equivalent to maximizing the Bhattacharyya distance in the equal-covariance case.

The dataset consists of 4 classes of 2D Gaussian distributions with the same covariance and different means (Figure 9A). Along the horizontal axis, the classes form two clusters that are far from each other, but that have high within-cluster overlap. Along the vertical axis, the classes are evenly separated at a short distance from each other, but with little overlap. The large distances along the horizontal axis will be highly weighted by the Bhattacharyya distance (i.e. LDA), whereas they will have a more modest contribution to the Fisher-Rao distance (Figure 9B). The filter learned by LDA aligns with the horizontal axis, whereas the first filter learned by SQFA aligns with the vertical axis (Figure 9A). The classes have less overlap along the vertical axis (Figure 9C), indicating that in this case, SQFA leads to better class separation than LDA. This example illustrates how the different distances can lead to different learning outcomes.

### D.2 Distances scale differently in the equal-mean case

Next, we consider the case of two Gaussians with zero mean and different covariances. In Figure 2 of the main text we showed how the distances scale for the 1D case. Here we complement that analysis by showing how the distances scale for the 2D case. We examine 2D distributions with diagonal covariance matrices, that is, $\Sigma_i = \mathbf{D} = \mathrm{diag}(\sigma_1^2, \sigma_2^2)$ and $\Sigma_j = \mathbf{I}$. Because of the affine invariance of all three distances, the general problem can be reduced to this special case.

Again, the three distances behave differently (Figure 10). The Hellinger distance is bounded between 0 and 1, while the Bhattacharyya and Fisher-Rao distances are unbounded. One interesting pattern is that whereas the contours of the Fisher-Rao distance are circles (as expected from its formula), the contours of the Hellinger and Bhattacharyya distances are more circular closer to the origin, but acquire a rhomboid shape farther from the origin.

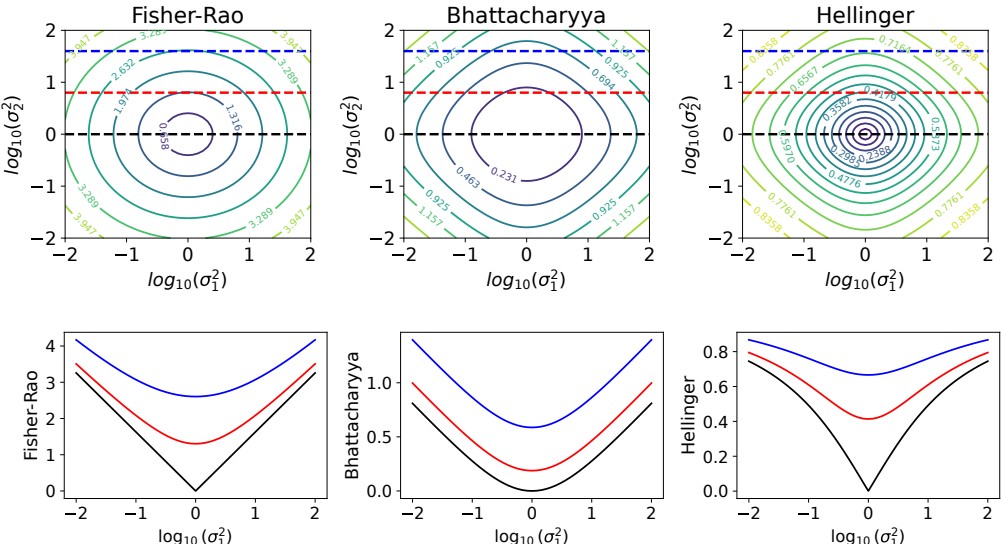

Figure 10: Distances as a function of covariance differences. The three distances (Fisher-Rao, Hellinger, Bhattacharyya) are shown as a function of $\log_{10}(\sigma_1^2)$ and $\log_{10}(\sigma_2^2)$, where the two Gaussian distributions are $\mathcal{N}(0, \mathbf{D})$ and $\mathcal{N}(0, \mathbf{I})$. The top row shows contour plots of the distances, while the bottom row shows 1D slices of the distances. The values used for the slices are shown as dashed lines in the top row.

The bottom plots show the three distances as a function of $\log_{10}(\sigma_1^2)$ for different fixed values of $\log_{10}(\sigma_2^2)$. This plot illustrates how the effect of one generalized eigenvalue depends on the value of the other generalized eigenvalue. We see that while the effect of $\log_{10}(\sigma_1^2)$ depends on the value of $\log_{10}(\sigma_2^2)$ for the Fisher-Rao and Hellinger distances, the two are independent for the Bhattacharyya distance. It is interesting to note that, of the different distances, the slices of the Fisher-Rao distance follow most closely the slices of the log-odds correct shown in Figure 7.

In sum, these examples illustrate some of the differences between the Fisher-Rao distance and other distances, in the special cases with closed-form expressions. The different behaviors of these distances mean that they will put different weights on the different pairwise distances between classes, leading to potentially different learning outcomes.

## E   Variability of results across runs

In this section we show the variability of the SQFA results across training runs.

In the experiments of the main text, except for the neural dataset, we ran each SQFA variant 10 times with different random seeds, and reported the median performance across runs. Here we report the variability across runs as the central 80% quantile range, that is, the difference between the 90% and the 10% quantiles across runs. To include the variability of the neural dataset, we used a new set of training runs, where we kept the neural recording session and train-test split fixed, and ran each SQFA variant 10 times on this single dataset. This way, the results below reflect SQFA training variability, not dataset variability.

We see that for most methods and datasets, the variability across runs is extremely low, of the order of 0.1% or less. For some conditions there is larger variability, especially when two filters are learned. For most cases, this variability comes from a few runs that found a poor local minimum, and so it does not change the overall pattern of results reported in the main text. The practical implication of this variability is that for most conditions random initialization does not seem to have a relevant effect on the final performance, but that for some conditions, especially when learning only two filters, it may be advisable to run the method multiple times with different random seeds and select the best-performing run.

Table 6: QDA accuracy 10–90% quantile range width over runs (as % correct).

| $m$ | Dataset | SQFA | smSQFA | SQFA-H | SQFA-B | SQFA-W | SQFA-J |
|---|---|---|---|---|---|---|---|
| 2 | SVHN | 3.5 | 4.4 | 6.7 | 3.9 | 0.1 | 0.0 |
| 4 | SVHN | 0.0 | 0.0 | 0.1 | 2.2 | 0.0 | 0.0 |
| 8 | SVHN | 2.5 | 0.1 | 0.0 | 4.0 | 0.0 | 0.0 |
| 16 | SVHN | 0.0 | 0.0 | 0.0 | 0.0 | 0.0 | 0.0 |
| 2 | MNIST | 8.2 | 4.1 | 2.6 | 0.2 | 0.6 | 0.0 |
| 4 | MNIST | 1.0 | 0.0 | 0.1 | 0.0 | 0.0 | 0.0 |
| 8 | MNIST | 0.0 | 0.1 | 0.1 | 0.4 | 0.0 | 0.2 |
| 16 | MNIST | 0.0 | 0.1 | 0.0 | 0.1 | 0.0 | 0.4 |
| 2 | Speed | 0.6 | 0.1 | 0.0 | 50.2 | 0.0 | 0.0 |
| 4 | Speed | 0.0 | 0.0 | 0.1 | 0.1 | 0.8 | 0.0 |
| 6 | Speed | 1.6 | 1.4 | 0.5 | 0.0 | 3.1 | 0.0 |
| 8 | Speed | 0.0 | 0.0 | 0.1 | 0.3 | 0.0 | 0.0 |
| 2 | Neural data | 0.0 | 0.0 | 0.0 | 0.0 | 0.0 | 0.0 |
| 4 | Neural data | 0.0 | 0.0 | 0.3 | 0.0 | 0.0 | 0.0 |
| 8 | Neural data | 2.2 | 0.0 | 0.6 | 0.0 | 0.0 | 0.0 |

## F  Robustness to the regularization parameter

In the main text, we selected the regularization parameter for the different SQFA variants using a validation set (except for the speed estimation dataset, where we used a parameter derived from the literature Burge & Geisler (2015)). Here we test the robustness of the method to the choice of the regularization parameter.

We learned 9 filters with SQFA, SQFA-H and SQFA-B using a range of values of the regularization parameter $\lambda$, for the SVHN and MNIST datasets, and 2 filters for the neural recordings dataset (we used only one of the neural recordings session, since different sessions have different optimal $\lambda$ values). All three methods showed a considerable degree of robustness to the choice of $\lambda$ (Figure 11). SQFA-H showed the highest robustness, with similar performance over a range of $\lambda$ values spanning an order of magnitude or more. SQFA and SQFA-B showed less robustness than SQFA-H, but still showed good performance over a large range of $\lambda$ values for all datasets. These results show that the performance, and the results in the main paper are not highly sensitive to the choice of the regularization parameter. For practical applications, it means that while selecting the regularization parameter using a validation set is advisable, this parameter does not require tight fine tuning.

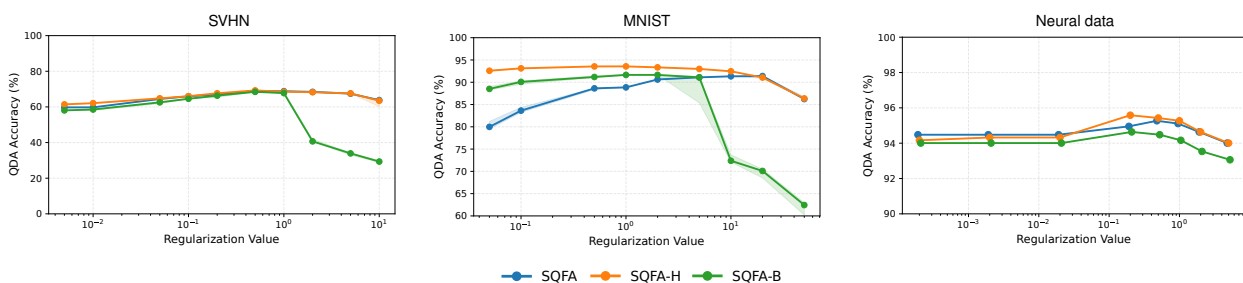

Figure 11:  Classification accuracy as a function of the regularization parameter. The accuracy of a QDA classifier trained on the features learned by SQFA, SQFA-H and SQFA-B is shown as a function of the regularization parameter $\lambda$, for the SVHN (left), MNIST (center), and neural recordings (right) datasets. The values of $\lambda$ are shown in log scale.

# G Complexity analysis and training times

## G.1 Computational complexity of SQFA

Here we analyze the computational complexity of SQFA. Specifically, we analyze the cost of taking a gradient step using the SQFA objective. For simplicity, we analyze the case of smSQFA (assuming zero means), but this is equivalent to the case of SQFA, since the operations are the same except substituting the $m \times m$ matrices with $(m+1) \times (m+1)$ matrices.

The first step is to compute the class covariances in the feature space. As mentioned in the Methods section, we achieve this by transforming the covariances of the raw data using the formula $\mathbf{\Sigma}_i = \mathbf{F}^T \mathbf{\Phi}_i \mathbf{F}$, where $\mathbf{\Phi}_i$ is the covariance of the data for class $i$. The matrix $\mathbf{\Phi}_i$ is $n \times n$, and the matrix $\mathbf{F}$ is $n \times m$, where $n$ is the dimensionality of the data and $m$ the number of filters. The cost of computing $\mathbf{\Sigma}_i$ for all $c$ classes is $O(cmn^2)$.

Next, we compute the generalized eigenvalues of the pair $(\mathbf{\Sigma}_i, \mathbf{\Sigma}_j)$, where the matrices are $m \times m$. The cost of computing $\mathbf{\Sigma}_i^{-1}$ is $O(m^3)$. Then, we compute the product matrix $\mathbf{\Sigma}_i^{-1}\mathbf{\Sigma}_j$, which also costs $O(m^3)$. Finally, the cost of computing the eigenvalues of $\mathbf{\Sigma}_i^{-1}\mathbf{\Sigma}_j$ is also $O(m^3)$. Performing this for all pairs of classes has a complexity of $O(c^2 m^3)$.

The cost of computing the gradient for the three operations described above is the same as the cost of computing the operations themselves. Thus, the complexity of taking a gradient step in SQFA is $O(c^2 m^3 + cmn^2)$.

It is important to note that the most expensive operations occur in the feature space, and the dimensionality of the feature space, $m$, is typically much smaller than the dimensionality of the data, $n$.

## G.2 Scaling analysis

Finally, we analyzed how SQFA scales with the data dimensionality and with the number of learned filters. We used five-class synthetic datasets in which all discriminative information was contained in a known informative subspace. The informative subspace was built by combining a subspace of dimension 4 with differences in the class means, and a complementary subspace where the classes had the same means but covariances with a minor and a major axis rotated with respect to one another. Thus, the informative subspace combined first- and second-order differences between classes. The rest of the dimensions had zero mean and identity covariance. The resulting space was rotated so that the informative subspace was not aligned with the canonical axes. We compared QDA accuracy using SQFA features against QDA accuracy using the true informative subspace.

For the first analysis, we generated different datasets, but for each we fixed the informative subspace to eight dimensions and embedded it in ambient spaces of increasing dimensionality. SQFA learned eight filters in each dataset. Across ambient dimensions up to 20,000, SQFA matched the accuracy obtained from the true informative subspace, with fit times ranging from seconds to minutes on a consumer laptop (Table 7).

Table 7: Scaling with ambient dimensionality. SQFA learned eight filters in a five-class synthetic problem with an eight-dimensional informative subspace.

| Ambient dimension | Max accuracy (%) | SQFA accuracy (%) | Fit time (s) |
|---|---|---|---|
| 1000 | 92.5 | 92.5 | 0.8 |
| 2000 | 92.5 | 92.6 | 2.6 |
| 5000 | 92.4 | 92.4 | 7.8 |
| 10000 | 92.7 | 92.6 | 30.7 |
| 20000 | 92.6 | 92.5 | 176.2 |

Second, we fixed the ambient dimensionality at 5000 and increased the dimension of the informative subspace. For each dataset, SQFA learned as many filters as there were informative dimensions. SQFA features achieved accuracy close to that obtained from the true informative subspace, including when learning hundreds of filters (Table 8).

Table 8: Scaling with the number of learned filters. The ambient dimensionality was fixed at 5000, and the number of filters matched the dimension of the informative subspace.

| $N$ filters | Max accuracy (%) | SQFA accuracy (%) | Fit time (s) |
|---|---|---|---|
| 10 | 65.9 | 65.9 | 10.8 |
| 50 | 76.0 | 73.2 | 13.8 |
| 100 | 77.5 | 74.6 | 22.2 |
| 500 | 64.9 | 63.7 | 85.0 |

Table 9: Training time (s).

| $m$ | Dataset | SQFA | smSQFA | SQFA-H | SQFA-B | SQFA-W | SQFA-J | LDA | SPCA | LFDA | WDA | LMNN | PCA |
|---|---|---|---|---|---|---|---|---|---|---|---|---|---|
| 8 | SVHN | 7.5 | 9.2 | 10.1 | 92.9 | 7.5 | 17.1 | 3.2 | 8.9 | 13.2 | 10.4 | 2340.0 | 1.9 |
| 8 | MNIST | 1.2 | 6.8 | 1.3 | 3.5 | 4.0 | 7.8 | 1.9 | 4.2 | 6.4 | 68.3 | 385.1 | 1.1 |
| 8 | Speed | 3.0 | 3.9 | 1.4 | 11.7 | 5.4 | 6.0 | 0.6 | 0.5 | 0.6 | 275.6 | 1936.8 | 1.2 |
| 8 | Neural data | 0.6 | 0.6 | 0.4 | 0.5 | 0.3 | 0.7 | – | 0.1 | 0.0 | 34.6 | 10.7 | 0.0 |

These examples show that SQFA can scale to tens of thousands of ambient dimensions and hundreds of filters in controlled synthetic settings. We note that here we used the known synthetic class statistics, but in real datasets, performance will also depend on how accurately the class statistics can be estimated. In the implementation used here, memory is eventually limited by storing dense class covariance matrices $\mathbf{\Phi}_i$; sparse covariance representations, optimization from raw data, or a preliminary PCA step could extend the practical scaling.

### G.3 Training times

For each dataset, we recorded the time it took to train the models on a consumer laptop with a 12th Gen Intel Core i7-1270P with 32 GB of RAM.

The training times of SQFA and its variants were in the order of seconds for all datasets (Figure 9). Remarkably, the training time for SQFA was comparable to that of the fastest training methods.

## H    Evaluation with kNN classifiers

In the main text, we evaluated the performance of the different methods by training a quadratic discriminant analysis (QDA) classifier on the learned features. Here, we show that the same patterns of results are obtained when using a k-nearest neighbor (kNN) classifier instead of QDA (Tables 10-13). All training and evaluation procedures were the same as in the main text, except that instead of training a QDA classifier, we trained a kNN classifier with $k = 5$.

Broadly, the results are similar to those obtained with QDA. For SVHN and MNIST, SQFA-H remained in general the best-performing method, and SQFA and SQFA-B also performed very well, among the top performing methods. It is remarkable that SQFA and SQFA-H remained highly competitive when using a kNN classifier, even when compared to methods that are considered to be better suited for kNN, such as LFDA, WDA and LMNN.

For the speed estimation dataset, there is some interesting effect of dimensionality, where the performance of kNN drops as the number of filters increases for most SQFA variants. The one exception is SQFA-J, but its special behavior can be explained by looking at its filters in Figure 4. The figure shows that SQFA-J learns coarsely the same pair of filters multiple times. For this dataset, this is known to happen when the learning procedure prioritizes minimizing the effects of internal noise (i.e. the regularization term) on a fixed pair of features rather than extracting new features Burge & Jaini (2017). Thus, by learning redundant features, the feature space of SQFA-J remains low-dimensional even as we increase the number of filters, avoiding the negative effects of increasing the dimensionality. Conversely, the methods that are not SQFA variants

have a low performance for 2 filters, but their performance increases with the number of filters, which is the expected behavior for these methods that are more suited for kNN.

Overall, these results show that the patterns of results obtained with QDA are not specific to that classifier, and that they generalize to a different type of classifier. This suggests that the SQFA variants should be a useful class of methods even when classifiers other than QDA are used.

Table 10: SVHN KNN classification accuracy (median percentage over runs).

| $m$ | SQFA | smSQFA | SQFA-H | SQFA-B | SQFA-W | SQFA-J | LDA | SPCA | LFDA | WDA | LMNN | PCA |
|---|---|---|---|---|---|---|---|---|---|---|---|---|
| 2 | **33.9** | 32.1 | **33.5** | 31.6 | 26.5 | 30.8 | 16.7 | 13.8 | 17.9 | 17.4 | 14.3 | 14.3 |
| 4 | **54.3** | 54.2 | **54.4** | 53.7 | 45.8 | 53.5 | 21.3 | 19.1 | 25.4 | 35.2 | 20.7 | 19.8 |
| 8 | **69.3** | **69.2** | **69.3** | 66.0 | 52.1 | 68.6 | 31.3 | 31.0 | 33.3 | 51.1 | 43.4 | 29.0 |
| 16 | **78.0** | 75.7 | **78.1** | 77.6 | 57.5 | 76.8 | – | 37.0 | 41.1 | 64.9 | 57.7 | 44.6 |

Table 11: MNIST KNN classification accuracy (median percentage over runs).

| $m$ | SQFA | smSQFA | SQFA-H | SQFA-B | SQFA-W | SQFA-J | LDA | SPCA | LFDA | WDA | LMNN | PCA |
|---|---|---|---|---|---|---|---|---|---|---|---|---|
| 2 | 57.2 | **57.7** | **62.9** | 51.9 | 48.4 | 45.9 | 52.3 | 44.2 | 54.9 | 50.5 | 47.2 | 42.4 |
| 4 | 79.9 | 77.1 | **86.7** | **82.6** | 74.3 | 76.1 | 82.0 | 71.3 | 81.9 | 67.2 | 73.2 | 63.3 |
| 8 | 91.8 | 90.8 | **93.9** | 92.1 | 91.4 | 89.9 | 92.0 | 90.9 | 90.8 | 90.1 | **92.7** | 90.1 |
| 16 | 96.0 | 96.2 | **96.6** | 96.1 | **96.3** | 94.2 | – | 94.9 | 93.7 | 96.1 | **96.3** | **96.3** |

Table 12: Speed KNN classification accuracy (median percentage over runs).

| $m$ | SQFA | smSQFA | SQFA-H | SQFA-B | SQFA-W | SQFA-J | AMA | LDA | SPCA | LFDA | WDA | LMNN | PCA |
|---|---|---|---|---|---|---|---|---|---|---|---|---|---|
| 2 | 53.0 | 53.3 | 51.3 | 53.1 | 30.7 | **53.9** | **56.3** | 4.3 | 7.0 | 4.7 | 6.2 | 3.4 | 23.3 |
| 4 | 29.8 | 29.5 | 30.4 | 28.4 | 23.3 | **59.7** | **36.0** | 8.9 | 18.1 | 17.1 | 14.3 | 10.6 | 19.8 |
| 6 | 22.0 | 23.4 | 25.6 | 20.8 | 21.8 | **61.7** | **29.1** | 15.7 | 17.2 | 8.0 | 28.3 | 17.0 | 16.2 |
| 8 | 23.5 | 23.4 | 26.6 | 18.7 | 20.4 | **63.3** | **29.6** | 19.5 | 17.1 | 25.2 | 29.0 | 22.0 | 17.4 |

# I    Regularization and invariance

## I.1    Invariance to invertible linear transformations

To better understand SQFA, it is important to consider the invariance properties of the Fisher-Rao and the affine-invariant distances.

First, we consider the affine-invariant distance and the case of smSQFA. If $\mathbf{G} \in GL(m)$ where $GL(m)$ is the General Linear group, composed of non-singular $m$-by-$m$ matrices, then

$$d_{AI}(\mathbf{\Psi}_i, \mathbf{\Psi}_j) = d_{AI}(\mathbf{G}^T \mathbf{\Psi}_i \mathbf{G}, \mathbf{G}^T \mathbf{\Psi}_j \mathbf{G}) \qquad (14)$$

In words, the affine-invariant distance is invariant to the action by congruence of $GL(m)$.

For a variable $\mathbf{z} \in \mathbb{R}^m$ with second-moment matrix $\mathbf{\Psi}_i$, the transformed matrix $\mathbf{\Psi}_i' = \mathbf{G}^T \mathbf{\Psi}_i \mathbf{G}$ corresponds to the second moment matrix of the transformed variable $\mathbf{z}' = \mathbf{G}^T \mathbf{z}$. Thus, the distance is invariant to invertible linear transformations of the underlying variable $\mathbf{z}$. Importantly, for the case of SQFA, where the variable $\mathbf{z}$ is obtained as $\mathbf{z} = \mathbf{F}^T \mathbf{x}$, this is also equivalent to a transformation of the filters. Specifically, if $\mathbf{z} = \mathbf{F}^T \mathbf{x}$, then $\mathbf{z}' = \mathbf{F}'^T \mathbf{x}$, where $\mathbf{F}' = \mathbf{FG}$.

In the context of smSQFA, and in the absence of regularization (see Methods section), if a set of filters are transformed as $\mathbf{F}' = \mathbf{FG}$, then the second-moment matrices for all the classes will be transformed as $\mathbf{\Psi}_i' = \mathbf{G}^T \mathbf{\Psi}_i \mathbf{G}$. Therefore, according to Equation 14, the pairwise distances between classes will be the same

Table 13: Zand-Kohn KNN classification accuracy (mean percentage over runs).

| $m$ | SQFA | smSQFA | SQFA-H | SQFA-B | SQFA-W | SQFA-J | LDA | SPCA | LFDA | WDA | LMNN | PCA |
|---|---|---|---|---|---|---|---|---|---|---|---|---|
| 2 | 89.4 | 86.9 | **90.3** | 86.9 | 84.6 | 85.6 | 87.2 | 84.6 | 86.9 | 86.2 | **90.2** | 65.3 |
| 4 | 93.7 | 94.4 | **95.6** | 92.4 | 84.7 | 91.4 | 93.0 | 91.9 | 93.1 | 89.1 | **95.0** | 90.6 |
| 6 | 95.8 | 96.0 | **96.4** | 95.9 | 84.6 | 95.1 | – | 96.1 | 96.4 | 84.3 | **96.2** | 95.6 |

for both sets of filters. In other words, the objective function of smSQFA is invariant to invertible linear transformations of the filters.

The Fisher-Rao distance and the Calvo-Oller bound are also invariant to affine transformations of the data space Nielsen (2020). This makes the filters learned by SQFA (like smSQFA above) non-unique up to invertible linear transformations (again, in the absence of regularization). To see this, let $\mathbf{H} = \begin{bmatrix} \mathbf{G} & \mathbf{0} \\ \mathbf{0} & 1 \end{bmatrix}$, where $\mathbf{0}$ is a $m$-by-1 vector of zeros, and $\mathbf{H} \in GL(m+1)$. We denote the moments of the transformed variable $\mathbf{z}' = \mathbf{G}^T \mathbf{z}$ as $\boldsymbol{\mu}' = \mathbf{G}^T \boldsymbol{\mu}$ and $\boldsymbol{\Sigma}'_i = \mathbf{G}^T \boldsymbol{\Sigma}_i \mathbf{G}$. Then we have the following relation

$$\mathbf{H}^T \boldsymbol{\Omega}_i \mathbf{H} = \mathbf{H}^T \begin{bmatrix} \boldsymbol{\Sigma}_i + \boldsymbol{\mu}\boldsymbol{\mu}^T & \boldsymbol{\mu} \\ \boldsymbol{\mu}^T & 1 \end{bmatrix} \mathbf{H} = \begin{bmatrix} \mathbf{G}^T(\boldsymbol{\Sigma}_i + \boldsymbol{\mu}\boldsymbol{\mu}^T)\mathbf{G} & \mathbf{G}^T\boldsymbol{\mu} \\ \mathbf{G}\boldsymbol{\mu}^T & 1 \end{bmatrix} = \begin{bmatrix} \boldsymbol{\Sigma}'_i + \boldsymbol{\mu}'\boldsymbol{\mu}'^T & \boldsymbol{\mu}' \\ \boldsymbol{\mu}'^T & 1 \end{bmatrix} = \boldsymbol{\Omega}'_i \quad (15)$$

Following the same reasoning as above, if the filters are transformed as $\mathbf{F}' = \mathbf{F}\mathbf{G}$, this amounts to transforming the Calvo-Oller embedding of each class as $\boldsymbol{\Omega}'_i = \mathbf{H}^T \boldsymbol{\Omega}_i \mathbf{H}$, meaning that the pairwise distances between classes remain the same.

In sum, in the absence of regularizing noise, there is an equivalent set of solutions, given by the sets of filters that span the same subspace. This can be a useful property, for example because there is no need to worry about scaling in the data space. But the lack of a unique solution also makes for less interpretable features (the interpretable object is the subspace spanned by the filters).

### I.2 Regularization breaks invariance

The equivalence of solutions above, however, is eliminated when we introduce regularization as an additive term $\mathbf{I}\sigma^2$ to the covariance matrices. We show this for the simpler case of smSQFA, but the same reasoning applies to SQFA.

As described in the Methods section, regularization is introduced by adding $\mathbf{I}\sigma^2$ to each second-moment matrix (equal to the covariance matrix in the zero-mean case of smSQFA) in the feature space. That is

$$\boldsymbol{\Psi}_i = \mathbf{F}^T \mathbb{E}\left[\mathbf{x}\mathbf{x}^T\right] \mathbf{F} + \mathbf{I}\sigma^2 \quad (16)$$

This means that if the filters are transformed as $\mathbf{F}' = \mathbf{F}\mathbf{G}$, it is no longer true that the regularized second-moment matrices are related by $\boldsymbol{\Psi}'_i = \mathbf{G}^T \boldsymbol{\Psi}_i \mathbf{G}$. Rather, there will be a different matrix $\mathbf{G}_i$ satisfying $\boldsymbol{\Psi}'_i = \mathbf{G}_i^T \boldsymbol{\Psi}_i \mathbf{G}_i$ for each class $i$. Therefore, the pairwise distances between classes will change, and the objective function will not be invariant to invertible linear transformations of the filters.

One consequence of adding regularization is that the solution is no longer invariant to the scale of the filters. For filters with small norms, matrices $\boldsymbol{\Psi}_i = \mathbf{F}^T \boldsymbol{\Phi}_i \mathbf{F} + \mathbf{I}\sigma^2$ will be dominated by the regularization term, and thus more similar (i.e. closer) to each other. Then, the solution will tend towards filters with infinite norm that make the contribution of the regularization term negligible. In our case the filters (i.e. each column of $\mathbf{F}$) are constrained to have unit norm, so this effect is avoided.

Following the same reasoning, the regularization term will penalize filters that lead to small second-moment matrices, since these will be dominated by the regularization term, which is identical across classes. Thus, regularization will favor the directions in the data space that lead to larger second-moment matrices. This might be useful in that it makes the filters more robust to estimation noise along dimensions with small variance, but it might also mask important information in directions with low squared values. Also, this makes the results dependent on the choice of the regularization parameter $\sigma^2$. Future work should explore

the effect of regularization on the features learned by SQFA, and examine how to choose the regularization parameter.

The breaking of invariance by regularization has the desirable effect of making the specific filters learned more reproducible and therefore more interpretable.

## J Details of the speed estimation task

### J.1 Dataset synthesis

The speed estimation dataset consists of synthetic naturalistic videos of surfaces moving at different frontoparallel speeds that were synthesized following the procedure described by Burge & Geisler (2015).

Briefly, each video is initially 60-by-60 pixels and 15 frames long. In this synthesis procedure, 60 pixels correspond to 1 degree of visual field (roughly equivalent to the foveal sampling of photoreceptors) and the videos have a duration of 250 ms (roughly equivalent to the duration of a fixation in natural viewing).

A video is synthesized by taking a random patch from an image of a natural scene and moving it horizontally at a given speed behind a 60-by-60 pixel aperture, for a duration of 15 frames. The resulting video is then filtered spatiotemporally to simulate the response of retinal photoreceptors to the video, following the procedure described by Herrera-Esposito & Burge (2024). Then, the resulting video is downsampled spatially by a factor of 2, to 30-by-30 pixel frames, and each frame is averaged vertically, leading to 30 pixel by 15 frame videos (thus, each video can be represented as a 450 dimensional vector). Vertically averaging the movies is equivalent to only considering filters that are vertically-oriented in the original 2D frame videos.

Then, to simulate further retinal processing, the video is converted to contrast, by subtracting and dividing by its mean intensity across pixels and frames. Denoting the resulting contrast video by $\mathbf{c}$, we apply divisive normalization to the video, dividing it by $\sqrt{(\|\mathbf{c}\|^2 + nc_{50}^2)}$ where $c_{50}^2 = 0.045$ is a constant and $n = 450$ is the number of pixels in the video multiplied by the number of frames. The resulting video simulates the retinal output in response to a naturalistic speed, as discussed in Burge & Geisler (2015); Herrera-Esposito & Burge (2024).

Above, we described the process by which a video is synthesized for a given speed. In the dataset, 41 retinal speeds (i.e. classes) were used, ranging from -6.0 to 6.0 deg/s with 0.3 deg/s intervals. For each of the 41 retinal speeds, we synthesized 800 different naturalistic videos, by using a different randomly sampled patch from a natural scene for each video. This adds nuisance naturalistic variability to the dataset. We used 500 videos per speed for training (a total of 20500 videos) and 300 videos per speed for testing (a total of 12300 videos).

Note that because the videos are generated with small patches randomly sampled from natural images, the expected intensity value at each pixel and frame is approximately the same (because natural image patches are approximately stationary), independent of speed. Then, because the videos were constrained to be contrast videos–i.e. they are formed by subtracting off and dividing by the intensity mean, consistent with early operations in the human visual system (Burge & Geisler, 2015)–the mean across patches equals zero, independent of speed. This results in a dataset where the classes are all approximately zero-mean.

### J.2 AMA-Gauss training

The AMA-Gauss model was implemented following the description in Jaini & Burge (2017); Herrera-Esposito & Burge (2024). First, a set of linear filters $\mathbf{F}$ is applied to each pre-processed video, and a sample of independent noise is added to each filter output. This results in a noisy response vector $\mathbf{R} = \mathbf{F}^T \mathbf{c} + \lambda$, where $\lambda \sim \mathcal{N}(0, \mathbf{I}\sigma^2)$ is the added noise. Then, a QDA-like decoder is used to classify the videos based on the noisy response vectors, assuming that the noisy response vectors are Gaussian distributed conditional on the speed (i.e. the class). The decoder computes the likelihood of the response given each class, that is, $p(\mathbf{R}|y = i)$, and using the priors (which we set as flat) and Bayes rule, it computes the posterior distribution $p(y|\mathbf{R})$ for the different speeds. For a given video $\mathbf{c}$ corresponding to true class $y = j$, the AMA-Gauss training loss is the negative log-posterior at the correct class, that is, $-\log p(y = j|\mathbf{R})$. The filters $\mathbf{F}$ that minimize the loss

are obtained using gradient descent. The filters (columns of $\mathbf{F}$) are constrained to have unit norm. We use the same regularization parameter $\sigma^2$ for AMA-Gauss and for SQFA. AMA-Gauss filters were learned in a pairwise fashion.

## K    Details of the neural decoding task

The response of a neuron in a given trial is the number of spikes that it produces in between 50 and 100 ms after stimulus onset (we used a fraction of the stimulus presentation time to avoid the performance ceiling). The recordings in the dataset were much longer, of 1.28 s, but we used a reduced window of 50-100 ms to make the task more challenging and to avoid ceiling effects.

We applied a preprocessing step to the dataset to remove outlier trials and neurons, because covariance estimation can be heavily affected by outliers and neural datasets can have a considerable number of outlier trials. We used a two-step procedure. First, we removed trials where the population response was an outlier. We standardized the population response for each class separately, and projected the resulting population response onto the first 10 principal components. We then obtained a robust covariance estimator using the Minimum Covariance Determinant (MCD) estimator. Lastly, we removed trials whose Mahalanobis distance to the class mean was larger than the median Mahalanobis distance plus 4 standard deviations. This procedure removed 1.2% of the trials.

Second, we removed neurons with a large number of outlier trials. For each neuron, we counted the number of trials with a firing rate deviating more than 3 standard deviations from the class mean. We removed neurons with more than 60 outlier trials. This procedure removed 3.3% of the neurons.

