# OpenReview forum: "Supervised Quadratic Feature Analysis: Information Geometry for Dimensionality Reduction"
_TMLR — Accepted by TMLR_

### Review · Reviewer_CMDY · 2026-01-22

**Summary Of Contributions:**

**Summary**

This work considers linear dimensionality reduction. Instead of classical methods like LDA or FDA, which try to "minimize intra-class variance and maximize between-class variance", the proposed method directly takes the information-theoretic viewpoint of maximizing a statistical distance between two labeled classes by taking covariance statistics of each class. The particular distance used is varied between the Fisher-Rao distance, Bhattacharyya and Hellinger distances. For the (uncomputable) Fisher-Rao variant, the authors propose to use a so-called Calvo-Oller lower bound as a statistical distance between Gaussians.

**Strengths**
* The interpretation of some supervised linear dimensionality reduction methods as optimizing some distance over the class-wise statistics is interesting.
* The proposed method seems novel at least.
* Figures are nice.

**Weaknesses/comments**

* This work could use a more detailed literature review, especially on existing methods in supervised (and unsupervised) linear dimensionality reduction. There are only two references in the text which do not appear to be discussed.
* What is discriminability? Is this a technical term? This is not clear in the current text and affects the clarity near the WDA area.
* Why can the filters be normalized to have unit norm? I am guessing it is because of the affine invariance but it is unclear in its current form. Please add a discussion on the affine invariance of the Fisher-Rao distance or Calvo-Oller bound.
* How is the Fisher-Rao distance / lower bound computed for the non-Gaussian experiments? If the data is parameterized using its sample mean/covariances, then the Wasserstein distance between the classes can also be easily computed in closed form.
* How does the proposed method work when the sample covariances between different classes are the same? Is it more/less sensitive than LDA? The zero-mean version smSQFA seems like it would not work.
* Is the Calvo-Oller bound only for zero-mean Gaussians, or does it also hold for more general distributions? The work claims that the Calvo-Oller bound can also hold for multivariate Student distributions etc, is there an analysis of using these distributions as priors instead?
* "Zero-mean Gaussians are relevant for some applications, and having an expression for the true distance in this special case is useful for validating the method". Please rewrite, the current form sounds uncomfortable. I am not sure where the method was validated using this either.
* The complete proposed method should be summarized somewhere. The current presentation in Section 2 is quite short yet does not clearly distinguish itself as the algorithmic part.
* Please compare with other supervised second order statistics methods. Notably, Fisher discriminant analysis is missing [Sugiyama07].
* The number of filters is super low. Some more recent works like [Vogelstein21] compare toy examples in millions of dimensions and hundreds of filters. Does the proposed method scale similarly? How does the method's feature usefulness scale with the number of filters?
* Can the proposed method be used with e.g. KL divergence between Gaussians? This has a closed form for Gaussians and the (I believe) the distances are also preserved under affine transformations.

Sugiyama, Masashi. "Dimensionality reduction of multimodal labeled data by local fisher discriminant analysis." _Journal of machine learning research_ 8.5 (2007).

Vogelstein, Joshua T., et al. "Supervised dimensionality reduction for big data." _Nature communications_ 12.1 (2021): 2872.

**Audience:**

Yes

**Audience Explanation:**

Relevant for compressed sensing audience.

**Claims And Evidence:**

No

**Claims Explanation:**

Insufficient experimental evidence, especially regarding scalability, ablations w.r.t number of filters, and comparisons with other methods. Insufficient explanations in the text regarding the approximations used and motivation behind the proposed method. No theoretical analysis present.

**Requested Changes:**

(Critical) Address discussed weaknesses.

---

> ### Author Response · Authors · 2026-04-09
>
> > **Strengths**
> >
> >- **The interpretation of some supervised linear dimensionality reduction methods as optimizing some distance over the class-wise statistics is interesting**
> >
> >- **The proposed method seems novel at least**
> >
> >- **Figures are nice**
>
> We thank the reviewer for their positive comments about the work.
>
> > **Weaknesses/comments**
> > - **This work could use a more detailed literature review, especially on existing methods in supervised (and unsupervised) linear dimensionality reduction. There are only two references in the text which do not appear to be discussed.**
>
> We agree that additional methods could be discussed. However, many dimensionality reduction methods exist, and only a small subset can be included. For comparison Vogelstein 2021, mentioned by the reviewer, does not mention most classes of methods discussed here.
>
> We also point out that we discussed more than "two references". Related Work discusses 1) LDA, 2) heteroscedastic LDA (Duin & Loog 2004, Choi & Lee 2003), 3) information-geometric LDA (Carter et.al 2009, Dwivedi et. al. 2022), 4) Wasserstein Discriminant Analysis (WDA, Flamary et. al. 2018), and 5) multi-class improvements of LDA (Thangavelu & Raich 2008). We also cited Supervised PCA (SPCA) (Barshan el al. 2011) and Large Margin Nearest Neighbors (LMNN) (Weinberger & Saul 2009) in the Methods (references in the main text).
>
> In the revised literature review we will mention unsupervised methods and have two separate sections: one about methods that maximize distances using class statistics (e.g. LDA and heteroscedastic LDA), and another about methods that use distances between samples (e.g. LFDA, WDA).
>
> > **What is discriminability? Is this a technical term? This is not clear in the current text and affects the clarity near the WDA area.**
>
> The term discriminability is commonly used in theoretical neuroscience and psychology (e.g. Dayan & Abbot 2005) to describe the statistical limits for detecting small stimulus perturbations, in the context of signal detection theory. It is theoretically linked to Fisher information. However, it is also used in a more colloquial way (e.g. Kriegeskorte & Wei 2021), analogous to the term “class separation”, which is how we use it here.
>
> In the revised manuscript, we will clarify our use of the term discriminability, and the specific passage flagged by the reviewer.
>
> > **Why can the filters be normalized to have unit norm? I am guessing it is because of the affine invariance but it is unclear in its current form. Please add a discussion on the affine invariance of the Fisher-Rao distance or Calvo-Oller bound.**
>
> Affine invariance is already discussed briefly in the Methods subsection titled "Regularization and invariance", and at length in the Appendix G.
>
> The Methods section says: "*Fisher-Rao distances are invariant to invertible linear transformations of the data, implying that learned filters are unique only up to the subspace they span.*". Same applies to the Calvo-Oller (CO) bound (Appendix G). This implies that normalization should not affect the results, as identified by the reviewer. We will state this explicitly in the revisions.
>
> However, in the same section we say that adding covariance regularization in the feature space breaks invariance during learning. Unit norm filters are required for regularization to work (Appendix G). The revisions will further discuss invariance in the Methods section, emphasizing the unit norm constraint.
>
> > **How is the Fisher-Rao distance / lower bound computed for the non-Gaussian experiments?**
>
> There is no Fisher-Rao (FR) distance computed for non-Gaussian distributions in this work. As stated in the Abstract, Introduction and Methods, SQFA as presented in this work operates "under Gaussian assumptions".
>
> We analyze the effects of non-Gaussianity in Appendix H, but focusing on the evaluation stage, not on the filter learning stage (which is not affected by non-Gaussianity, because it relies only on summary statistics). We will make this point clearer in the text.
>
> > **If the data is parameterized using its sample mean/covariances, then the Wasserstein distance between the classes can also be easily computed in closed form.**
>
> The reviewer is correct. We performed this analysis, and found that the Wasserstein distance (SQFA-W) performs worse than the FR distance (SQFA). We also tested two other methods suggested by the reviewer below: SQFA with the Jeffreys divergence (SQFA-J), and LFDA. Both performed worse than SQFA. **Table 1** shows the median QDA accuracy (5 repetitions) obtained with each set of features:
>
> **Table 1**
> | Dataset | SQFA | SQFA-W | SQFA-J | LFDA |
> |-|-|-|-|-|
> | SVHN | 68.8% | 61.5%| 68.8%| 35.5% |
> | MNIST| 91.4% | 90.1%| 90.1%| 90.2% |
> | Speed| 88.1% | 61.5%| 74.3%| 51.1% |
>
> We will add these results in the revised version.
>
> It is worth mentioning that SQFA-W is different from WDA [Flamary 2018], although the revisions will also include WDA in response to reviewer bDFH.

---

> > ### Author Response · Authors · 2026-04-09
> >
> > > **How does the proposed method work when the sample covariances between different classes are the same? Is it more/less sensitive than LDA? The zero-mean version smSQFA seems like it would not work.**
> >
> > We thank the reviewer for raising this useful question. When the sample covariances are the same, our method will work similarly, but not identically to LDA.
> >
> > LDA filters maximize the sum of squared Mahalanobis distances between classes (Appendix D). For two homoscedastic Gaussians, the FR distance is a sub-linear monotonic function of the Mahalanobis distance (Appendix C2, **Figure 10**). Then, for a pair of classes, maximizing the squared Mahalanobis distance is equivalent to maximizing the FR distance.
> >
> > But when there are more than two classes, the objectives of SQFA and LDA can differ. In this case, there is a tradeoff between maximizing the distances of the different pairs. Because the FR distance scales differently than the squared Mahalanobis distance, SQFA and LDA will place different weights on the different pairs.
> >
> > To illustrate this point, we designed a toy dataset in which the first LDA feature fails at recovering the subspace with best separation (similar to [Tao 2007]), and where the first SQFA feature yields better performance. The **Figure 1** with the example is available at the anonymized link https://osf.io/ytc3g/overview?view_only=bea3890103f7440da60c9e8764a20448.
> >
> > The dataset has 4 classes with equal covariance in a 2D data space (**Figure 1A**). Along the horizontal axis, the classes form two distant clusters, with high within-cluster overlap. Along the vertical axis, classes are evenly separated, with little overlap but smaller distances. The large horizontal separation leads to large squared Mahalanobis distances (LDA’s objective), but to moderate FR distances (SQFA’s objective). When learning a single filter, LDA aligns with dimension #1 (horizontal axis), whereas SQFA aligns with the more discriminative dimension #2 (vertical axis) (**Figure 1A**). **Figure 1B** shows the classes projected onto LDA (top) and SQFA (bottom) filters.
> >
> > **Figure 1C** further illustrates this, showing how the relative difference between the between-cluster distance (green dot) and the between-class distance (magenta) is larger for the LDA objective (orange curve) than for the SQFA objective (blue curve). This example illustrates that, overall, SQFA should be less sensitive to outlier classes.
> >
> > These issues are already discussed in Appendix C (**Figure 10**), but this new illustrative example will be included in the revised Appendix and mentioned in the main text.
> >
> > > **Is the Calvo-Oller bound only for zero-mean Gaussians, or does it also hold for more general distributions? The work claims that the Calvo-Oller bound can also hold for multivariate Student distributions etc, is there an analysis of using these distributions as priors instead?**
> >
> > We note an apparent confusion in this comment. The zero-mean Gaussian case has an exact closed form expression for the FR distance (**Equation 4**), and so the bound is not needed in that case (it actually equals the exact distance, see Appendix C). The CO bound is used for non- zero-mean Gaussians. We will make this clearer in the revisions.
> >
> > Regarding non-Gaussian distributions, the FR distance is the same for all zero-mean elliptical distributions. So, when all classes are zero-mean (section 5.4), SQFA is agnostic about what specific elliptical distribution the data follows.
> >
> > For the non-zero-mean case, we use the CO bound under Gaussian assumptions. CO bounds are available for non-Gaussian elliptical distributions (e.g. the Student’s distribution), but the bound for each distribution is given by a different formula. Future work could make use of such formulas to learn filters under different distributional assumptions.
> >
> > > **"Zero-mean Gaussians are relevant for some applications, and having an expression for the true distance in this special case is useful for validating the method". Please rewrite, the current form sounds uncomfortable. I am not sure where the method was validated using this either.**
> >
> > We agree that this sentence is awkward and requires rewriting.
> >
> > As emphasized in the previous response, there is an exact formula for the FR distance between zero-mean Gaussians. For arbitrary Gaussians, we use the CO bound instead. One issue when using the CO bound is that it is unclear to what extent the results would be different if the true FR distance could be used. In the zero-mean case the use of an approximation is not a problem, and the learned filters reflect the class separation achieved by the true FR distance. This provides complementary insights to the non-zero mean case, which is put to use in section 5.4, where we analyze a video dataset with zero mean classes.
> >
> > We will clarify the sentence above, and better explain this point in section 5.4.

---

> ### Author Response · Authors · 2026-04-09
>
> > **The complete proposed method should be summarized somewhere. The current presentation in Section 2 is quite short yet does not clearly distinguish itself as the algorithmic part.**
>
> See below a high-level description of the algorithm. We will include a similar description in the revised manuscript.
>
> **Inputs:**
> - Class means $\\{\gamma_i\\}_{i=1}^c$
> - Class covariances $\\{\Phi_i\\}_{i=1}^c$
> - Number of filters $m$
> - Regularization $\sigma^2$
>
> **Output:** filters $F \in \mathbb{R}^{n \times m}$
>
> 1. Initialize unconstrained parameters $\eta \in \mathbb{R}^{n \times m}$.
> 2. Set $\mathcal{L}_{\mathrm{prev}} \gets -\infty$ and $\Delta \gets \infty$.
> 4. While $\Delta \ge 10^{-6}$:
>    1. Compute the constrained filters from the unconstrained parameters:
>       $$
>       F \gets g(\eta),
>       $$
>       where $g$ normalizes each column of $\eta$.
>    2. For each $i = 1,\dots,c$:
>       $$
>       \mu_i \gets F^\top \gamma_i
>       $$
>       $$
>       \Sigma_i \gets F^\top \Phi_i F + \sigma^2 I_m
>       $$
>       $$
>       \Omega_i \gets
>       \begin{bmatrix}
>       \Sigma_i + \mu_i \mu_i^\top & \mu_i \\\\
>       \mu_i^\top & 1
>       \end{bmatrix}
>       $$
>    3. For each $i = 1,\dots,c$ and $j = 1,\dots,c$:
>       $$
>       D_{ij} \gets d_{\mathrm{AI}}(\Omega_i,\Omega_j)
>       $$
>    4. Compute
>       $$
>       \mathcal{L}(\eta) \gets \sum_{i<j} D_{ij}.
>       $$
>    5. Update $\eta$ using one L-BFGS step to maximize $\mathcal{L}(\eta)$.
>    6. Compute
>       $$
>       \Delta \gets \left| \mathcal{L}(\eta) - \mathcal{L}_{\mathrm{prev}} \right|.
>       $$
>    7. Set
>       $$
>       \mathcal{L}_{\mathrm{prev}} \gets \mathcal{L}(\eta).
>       $$
> 5. Return
>    $$
>    F = g(\eta).
>    $$
>
> > **Please compare with other supervised second order statistics methods. Notably, Local Fisher discriminant analysis is missing [Sugiyama07].**
>
> We did this analysis and found that SQFA outperforms LFDA in all tested datasets. We used the LFDA implementation of the Python package metric-learn, and cross-validated its number-of-neighbors hyperparameter. The revised manuscript will include these results.
>
> We note two related points. First, the initial submission included other second-order methods. Specifically, SQFA-B is analogous to a method sometimes called Heteroscedastic LDA (Duin & Loog 2004), and Supervised PCA also makes use of second-order information. We will make this clearer in the revisions.
>
> Second, we think that it is more accurate to describe LFDA not as a second-order method, but as a metric-learning method, like LMNN which is already included in the paper. For example, LFDA is especially useful when classes are multimodal.

---

> > ### Author Response · Authors · 2026-04-09
> >
> > > **The number of filters is super low. Some more recent works like [Vogelstein21] compare toy examples in millions of dimensions and hundreds of filters. Does the proposed method scale similarly? How does the method's feature usefulness scale with the number of filters?**
> >
> > We agree that this is an interesting venue to explore. First, we would like to note that Vogelstein 21 is probably not a good comparison standard. Their method is a combination of LDA and class-centered PCA. More computationally complex methods like WDA or LMNN have more modest scaling. Also, Vogelstein 21 focuses on features for linear classifiers, while SQFA focuses on quadratic classifiers. Quadratic classifiers extract more information per feature but also overfit more easily (which SQFA can help address). Thus, learning thousands of features is less desirable for SQFA.
> >
> > That said, we developed two toy problems to study the scalability of our method. In both examples below we used 5 classes.
> >
> > In the first toy problem, we studied how SQFA scales with the data dimensionality. For this, we learned 8 filters while scaling the data dimensionality. We generated class statistics such that an 8-dimensional subspace in the data space had all informative differences between the classes, with differences in means and covariances like the toy problems in **Figure 2** of the submitted paper. This subspace was embedded in an ambient space of increasing dimensionality (up to $2\times 10^4$), where all other dimensions were uninformative (the informative subspace was not aligned with the canonical axes). We learned 8 filters with SQFA on these datasets, and computed QDA accuracy using both the feature space learned by SQFA, and the true 8-dimensional informative subspace (maximum performance). SQFA features achieved maximum performance for all embedding dimensions (**Table 2** below), and training completed over seconds/minutes on a consumer laptop. For scaling the ambient dimensionality further, memory requirements become limiting, because of large dense data covariance matrices (i.e. $\Phi_i$). Simple computational modifications like using sparse data covariances, using the raw data and not the covariances for optimization, or performing a pre-processing PCA could allow scaling to proceed further.
> >
> > **Table 2**
> > |Ambient dim| Max accuracy (%) |SQFA accuracy (%)	| Fit time (s) |
> > |-|-|-|-|
> > |1000	| 92.5| 92.5| 0.8|
> > |2000	| 92.5| 92.6| 2.6|
> > |5000	| 92.4| 92.4| 7.8|
> > |10000| 92.7| 92.6| 30.7|
> > |20000| 92.6| 92.5| 176.2|
> >
> > In the second toy problem, we studied how SQFA scales with the number of filters learned. The setup was similar to the problem above, but we fixed the data dimension at $5000$, and increased the informative subspace dimensionality (up to $500$). For each dataset we learned a number of filters equal to the informative dimensions, and computed QDA accuracy using the SQFA features and the true informative subspace. SQFA features achieved close to maximum performance in all cases (**Table 3** below), and were trained in seconds on a consumer laptop.
> >
> > **Table 3**
> > |N filters| Max acc (%)	|SQFA acc (%)	|Fit time (s)
> > |-|-|-|-|
> > |10	|65.9	|65.9|10.8|
> > |50	|76.0	|73.2|13.8|
> > |100|77.5 |74.6|22.2|
> > |500|64.9	|63.7|85|
> >
> > These examples above show that SQFA can easily scale to many thousands of dimensions and hundreds of filters. A caveat is that we used the true data statistics, and in a real-world high-dimensional problem performance will depend on the noisiness of the estimated data statistics (which is true for any similar method). We will include this analysis in the Appendix.
> >
> > Additionally, see the response to reviewer bDFH for an additional analysis varying the number of filters from 2 to 20 for all analyses in the paper.
> >
> > > **Can the proposed method be used with e.g. KL divergence between Gaussians? This has a closed form for Gaussians and the (I believe) the distances are also preserved under affine transformations.**
> >
> > We ran this analysis using the Jeffreys divergence (symmetrized KL) with SQFA (SQFA-J), which performed worse than SQFA (**Table 1**). These results will be included in the revisions.
> >
> > **References**
> >
> > Dayan, P., & Abbott, L. F. (2005). Theoretical neuroscience: computational and mathematical modeling of neural systems. MIT press.
> >
> > Flamary, Rémi, et al. "Wasserstein discriminant analysis." Machine Learning 107.12 (2018): 1923-1945.
> >
> > Kriegeskorte, N., & Wei, X. X. (2021). Neural tuning and representational geometry. Nature Reviews Neuroscience, 22(11), 703-718.
> >
> > Sugiyama, Masashi. "Dimensionality reduction of multimodal labeled data by local fisher discriminant analysis." Journal of machine learning research 8.5 (2007).
> >
> > Tao, Dacheng, et al. "General averaged divergence analysis." Seventh IEEE International Conference on Data Mining (ICDM 2007). IEEE, 2007.
> >
> > Vogelstein, Joshua T., et al. "Supervised dimensionality reduction for big data." Nature communications 12.1 (2021): 2872.

---

### Review · Reviewer_bDFH · 2026-02-09

**Summary Of Contributions:**

The paper proposes a supervised linear dimensionality-reduction method that learns a projection by maximizing the Fisher–Rao distance between class-conditional Gaussians in the projected space.

They first point out a nice special case: with zero mean, the second-moment version (“smSQFA”) boils down to an affine-invariant distance on SPD matrices. Then for the general (non-zero mean) case, they approximate Fisher–Rao distance between multivariate Gaussians using the Calvo–Oller construction.

They optimize the objective with L-BFGS and mostly evaluate with QDA on the learned features. They report solid improvements over common supervised/unsupervised DR baselines on digit datasets, and competitive results on a multi-class speed-estimation task, where the learned filters also look interpretable.

Strengths
* The method is explained clearly.
* They make it tractable (exact special case + workable approximate), and the comparisons to Bhattacharyya/Hellinger feel fair since they keep the optimizer the same.
* The results match their story: it can use both mean and covariance differences.
* The speed-estimation task is a nice qualitative check (filters aren’t just random-looking).

Weaknesses
* The “why Fisher–Rao?” part still doesn’t land for me. They explain why it’s a reasonable discriminability distance, but not the intuition for why it should beat other existing distances. So when it wins, it can feel a bit like luck rather than “the theory predicts this.”
* They lean a lot on QDA, which creates a bit of a closed loop: optimize Gaussian class distances, then evaluate with a Gaussian classifier.
* Baselines: OT/Wasserstein-style discriminant analysis is basically replaced by LMNN, which isn’t really the same thing.

**Audience:**

Yes

**Audience Explanation:**

* The paper is relevant to readers interested in supervised dimensionality reduction or information geometry, since it gives a concrete, workable example of using Fisher–Rao ideas in an optimization-based ML method.
* It also provides a reasonable baseline for comparing different Gaussian-distance objectives under the same optimization setup.

**Broader Impact Concerns:**

No concern.

**Claims And Evidence:**

Yes

**Claims Explanation:**

The paper defines a concrete supervised linear DR objective based on Fisher–Rao separation of class-conditional Gaussians. It then explains how it becomes tractable via an exact special case (zero-mean) and a practical approximation, and uses a standard optimizer with a clear training procedure.

The empirical evidence is generally consistent with the paper’s intended use case which is quadratic discrimination. The toy examples illustrate the mean-vs-covariance behavior they claim. The digit datasets show consistent gains over common dimentionality reduction baselines. The authors swap Fisher–Rao with Bhattacharyya/Hellinger while keeping the optimizer fixed, which makes the comparisons convincing.

Again, the weaker part is the motivation/justification: the paper doesn’t fully explain why Fisher–Rao should be expected to beat other Gaussian distances in general, and it heavily relies on QDA so the evaluation feel somewhat self-contained -- optimize Gaussian separation, evaluate with a Gaussian decoder.

**Requested Changes:**

* Again, in addition to QDA, we need to evaluate the learned features with at least one simple alternative (e.g., linear classifier / logistic regression / nearest-center). This would reduce the closed-loop feel and clarify how much of the gain is specific to the Gaussian decoder.
* If possible, please provide a clearer reason to expect Fisher–Rao to be a good objective for DR, and explicitly contrast this with why Bhattacharyya/Chernoff/Hellinger might or might not be preferable.
* Using LMNN as a proxy is not fully satisfying. Please either add a closer implementation/baseline (even if approximate), or clearly re-scope the claim and discuss the gap more explicitly.
* Ablations / sensitivity on key hyperparameters are missing, such as projection dimension and regularization strength.
* It would be great to provide more explicit guidance on initialization, convergence behavior, and runtime scaling in practice. Consider moving a few key points from Appendix B/C/H into the main paper, e.g., (i) the main intuition from Appendix B for the zero-mean case, (ii) a compact quantitative check from Appendix C showing how faithful the Calvo–Oller surrogate is in practice, and (iii) the main takeaway from Appendix H on what happens when Gaussianity is violated.

---

> ### Author Response · Authors · 2026-04-09
>
> > **Strengths**
> >
> >- **The method is explained clearly.**
> >
> >- **They make it tractable (exact special case + workable approximate), and the comparisons to Bhattacharyya/Hellinger feel fair since they keep the optimizer the same.**
> >
> >- **The results match their story: it can use both mean and covariance differences.**
> >
> >- **The speed-estimation task is a nice qualitative check (filters aren’t just random-looking).**
>
> We thank the reviewer for their positive comments.
>
> > **Again, in addition to QDA, we need to evaluate the learned features with at least one simple alternative (e.g., linear classifier / logistic regression / nearest-center). This would reduce the closed-loop feel and clarify how much of the gain is specific to the Gaussian decoder.**
>
> Unless we misunderstand the reviewer’s remark, the requested analysis was presented in Appendix H of the original submission  (“Effect of Gaussianity of model evaluation”; see **Figure 14**), where we test the accuracy of a KNN classifier using the filters learned by different methods. We realize that this should have been mentioned more emphatically in the main text. Furthermore, the reviewer will see below further results using KNN when varying the number of learned filters (**Tables 4,5,6**), which we may include in the updated manuscript. SQFA filters still perform remarkably well with the KNN classifiers.
>
> A revised version of the manuscript can either keep the evaluation with KNN classifier in the Appendix and expand their discussion in the main text, or directly show results with KNN classifier in the main text.
>
> We are inclined to keep the KNN results in the Appendix, because we believe that using the QDA decoder should not be interpreted as a “closed-loop” that hinders interpretation. SQFA has the goal of maximizing the differences between classes under Gaussian assumptions, and the best way to evaluate whether it succeeds is by testing it with a classifier that makes the same assumptions. Another way to think about it is that the goal of SQFA is to find features that lead to the best QDA classification. Thus, we think that evaluating SQFA filters with other classifiers, while important, can also be misleading, and break the narrative. As an example, the LMNN method tries to maximize kNN classification, and the paper introducing the method focuses on this classifier (Weinberger 2009) without testing linear or quadratic classifiers. These considerations are discussed and analyzed at length in Appendix H.
>
> In the main text, we will more prominently indicate the issues discussed in this Appendix section. If the reviewer still thinks it is essential that the KNN results are included in the main text, we are open to doing so if space constraints allow.
>
> > **If possible, please provide a clearer reason to expect Fisher–Rao to be a good objective for DR, and explicitly contrast this with why Bhattacharyya/Chernoff/Hellinger might or might not be preferable.**
>
> The Fisher-Rao distance is a natural objective if we adopt a geometric view of the dimensionality reduction problem as maximizing a distance between classes, because it is the canonical Riemannian distance in information geometry. It being a geodesic distance gives it an appealing geometric intuition as an objective function. Furthermore, because of the importance of the Fisher-Rao distance in some machine learning and neuroscience applications, it is an important question how the goal of maximizing these distances relates to machine learning objectives like classification accuracy.
>
> Despite the above, it seems to us that there is no obvious ironclad theoretical reason to think that the Fisher-Rao distance should maximize classification accuracy when used as a dimensionality reduction objective. In the Section 2 of the paper, we reason that the Fisher-Rao distance "*can be conceptualized as the accumulated discriminability of the infinitesimal perturbations transforming $\mathcal{N}(\mu_i, \Sigma_i)$ into $\mathcal{N}(\mu_j, \Sigma_j)$ along the geodesic*”, which is suggestive of the distance being a good objective that maximizes classification accuracy, but not a rigorous proof. In Section 3: Related work, we also write: "On the other hand, the distances $d_B$ (Bhattacharyya) and $d_H$ (Hellinger) have direct links to Bayes error in the two-class case, unlike $d_{FR}$ (Fisher-Rao).".
>
> Thus, in lack of a strong theoretical reason, this work contributes an empirical evaluation of Fisher-Rao distance maximization on complex real-world datasets, enabled by the use of the Calvo-Oller bound. We hope that the present work will motivate further theoretical work analyzing the Fisher-Rao distance as a dimensionality reduction objective. We will make these points clearer in the revised manuscript.

---

> > ### Comment · Reviewer_bDFH · 2026-04-15
> > **Why Fisher–Rao**
> >
> > I agree that there isn’t an ironclad theory that Fisher–Rao should optimize classification accuracy.
> >
> > Given that, I’d suggest making the framing in the main paper a bit more explicit like this is a principled information-geometry objective plus a practical surrogate (Calvo–Oller) plus an empirical evaluation showing it’s competitive/strong in practice, rather than implying Fisher–Rao is expected to be superior.
> >
> > Also, the contrast with Bhattacharyya/Hellinger would be stronger if we can make it more concrete: if it’s not about Bayes-error bounds, then what’s the practical reason that someone should use Fisher–Rao? (invariance, stability, multi-class weighting, optimization properties, etc.)

---

> ### Author Response · Authors · 2026-04-09
>
> >**Using LMNN as a proxy is not fully satisfying. Please either add a closer implementation/baseline (even if approximate), or clearly re-scope the claim and discuss the gap more explicitly.**
>
> To address this remark, we built our own implementation of WDA, combining Python’s package POT optimal transport functionalities with PyTorch’s efficient optimization algorithms. We will make this more computationally efficient implementation of WDA available upon publication, which should be a useful tool for the community.
>
> SQFA performs in general better than WDA for the datasets used here, as evaluated with both QDA and KNN accuracy (see **Tables 1-6** below). Like for SQFA, we used cross-validation to select the best regularization parameter of WDA. Additionally, our results for classification performance for MNIST after WDA dimensionality reduction are in agreement with those of the original paper (Flamary et al.), with classification performance accuracy around 90% when learning 10 filters.
>
> We thank the reviewer for motivating this additional analysis. We believe it makes our paper a more comprehensive comparison across state-of-the-art methods.
>
> > **Ablations / sensitivity on key hyperparameters are missing, such as projection dimension and regularization strength.**
>
> We thank the reviewer for this suggestion. We performed both sensitivity analyses.
>
> First, we evaluated all methods across a range of feature dimensions. We ranged from 2 to 20 features for SVHN and MNIST, and from 2 to 8 for the speed estimation dataset, following the original publication. For all SQFA variants, WDA, LFDA, and LDA, we cross-validated the regularization parameter for each feature dimension. We show in bold the two highest scores for each feature dimension.
>
> **Table 1** SVHN, QDA accuracy (%)
> |Nfilters|SQFA|SQFA-H|SQFA-B|LDA|SPCA|LFDA|WDA|LMNN|PCA|
> |--|--|--|--|--|--|--|--|--|--|
> |2|**39.9**|38.3|**38.6**|23.6|19.6|22.9|22.3|19.5|19.6|
> |4|**56.4**|**56.9**|55.9|27.4|22.8|30.6|39.9|25.9|24.5|
> |8|**68.2**|**68.2**|**67.3**|34.8|37.5|33.5|57.1|49.9|35.6|
> |12|**73.0**|**72.8**|72.6||45.6|28.4|63.5|60.4|50.5|
> |16|**74.7**|**74.8**|74.4||50.4|34.4|66.4|67.5|58.4|
> |20|**76.4**|**76.3**|**76.3**||55.7|35.6|67.8|70.0|63.5|
>
> **Table 2** MNIST, QDA accuracy (%)
> |Nfilters|SQFA|SQFA-H|SQFA-B|LDA|SPCA|LFDA|WDA|LMNN|PCA|
> |--|--|--|--|--|--|--|--|--|--|
> |2|**64.4**|**67.1**|56.0|56.6|48.4|59.2|55.1|52.6|46.1|
> |4|79.7|**86.6**|**82.5**|**82.5**|72.9|82.2|68.9|73.5|63.1|
> |8|89.7|**93.2**|**90.8**|90.1|88.5|89.4|86.9|90.2|86.5|
> |12|92.8|**94.9**|**92.9**||90.9|90.5|91.1|**92.9**|91.7|
> |16|94.2|**95.4**|**94.4**||92.5|90.5|93.8|94.2|93.7|
> |20|95.0|**95.5**|94.7||93.8|90.5|94.7|**95.1**|95.0|
>
> **Table 3** Speed estimation, QDA accuracy (%)
> |Nfilters|SQFA|SQFA-H|SQFA-B|SPCA|LDA|LFDA|WDA|LMNN|PCA|
> |--|--|--|--|--|--|--|--|--|--|
> |2|**58.0**|56.0|**58.2**|9.5|5.4|4.9|20.2|4.7|23.9|
> |4|**68.0**|**67.4**|66.3|31.3|11.2|25.3|23.8|15.3|32.8|
> |6|**79.1**|**83.5**|78.3|40.3|20.9|11.5|31.0|35.3|48.7|
> |8|**89.3**|**89.2**|85.0|59.8|28.1|52.8|38.9|51.0|75.1|
>
> Next we show the same filters evaluated with KNN
>
> **Table 4** SVHN, KNN accuracy (%)
> |Nfilters|SQFA|SQFA-H|SQFA-B|LDA|SPCA|LFDA|WDA|LMNN|PCA|
> |--|--|--|--|--|--|--|--|--|--|
> |2|**34.1**|**33.2**|32.8|16.7|13.8|17.9|17.4|14.3|14.3|
> |4|**54.2**|**54.5**|53.7|21.3|19.1|25.4|35.2|20.7|19.8|
> |8|**69.5**|**69.4**|68.5|31.3|31.0|33.3|51.1|43.4|29.0|
> |12|74.6|**76.2**|**75.8**||36.1|26.4|62.6|51.8|39.7|
> |16|**78.0**|**78.1**|77.6||37.0|41.1|65.4|57.7|44.6|
> |20|**78.6**|**79.3**|78.5||38.9|41.2|66.6|59.1|47.0|
>
> **Table 5** MNIST, KNN accuracy (%)
> |n_filters|SQFA|SQFA-H|SQFA-B|LDA|SPCA|LFDA|WDA|LMNN|PCA|
> |--|--|--|--|--|--|--|--|--|--|
> |2|**60.9**|**62.9**|52.1|52.3|44.2|54.9|50.6|47.2|42.4|
> |4|79.7|**86.7**|**82.6**|82.0|71.3|81.9|68.2|73.2|63.3|
> |8|91.8|**93.9**|92.1|92.0|90.9|90.8|90.3|**92.7**|90.1|
> |12|94.7|**95.8**|94.7||93.6|93.1|94.4|**95.2**|94.4|
> |16|96.0|**96.6**|96.1||94.9|93.7|96.1|**96.3**|**96.3**|
> |20|96.7|**96.8**|96.6||95.7|93.7|96.6|**96.9**|**96.9**|
>
> **Table 6** Speed estimation, KNN accuracy (%)
> |n_filters|SQFA|SQFA-H|SQFA-B|LDA|SPCA|LFDA|WDA|LMNN|PCA|
> |--|--|--|--|--|--|--|--|--|--|
> |2|**53.1**|**51.4**|**53.1**|4.3|7.0|4.7|17.7|3.4|23.3|
> |4|29.8|**30.5**|**31.1**|8.9|18.1|17.1|19.0|10.6|19.8|
> |6|23.4|**25.9**|20.9|15.7|17.2|8.0|**28.9**|17.0|16.2|
> |8|23.6|**26.5**|19.0|19.5|17.1|25.2|**29.2**|22.0|17.4|
>
> The results show that, across datasets and feature dimensions, SQFA-H generally outperforms other methods for both QDA (Figure 1) and KNN accuracy (Figure 2). SQFA typically is among the highest-performing methods across the number of feature dimensions. This highlights the potential value of the methods introduced in this work for dimensionality reduction, and the utility of learning low-dimensional feature spaces that allow for good classification accuracy. If space allows, we might show these results in the main text.

---

> > ### Author Response · Authors · 2026-04-09
> >
> > Second, we tested the effect of the regularization parameter on performance for SVHN and MNIST (since the motion dataset uses a fixed regularization value informed by neuroscience). We found that SQFA-H was the most robust to the regularization value, followed by SQFA and lastly by SQFA-B. For SQFA, changes of 1 order of magnitude in the regularization parameter around the optimal value had little effect on performance. Larger changes reduced performance gradually. Hence, the results are robust to the value of this hyperparameter. The analysis of regularization sensitivity will be added to the Appendix.
> >
> > **Table 7** SVHN, QDA accuracy vs regularization.
> > |regularization|SQFA|SQFA-H|SQFA-B|
> > |--|--|--|--|
> > |0.005|59.9|61.4|58.1|
> > |0.01|59.8|62.1|58.8|
> > |0.05|64.5|65.0|62.7|
> > |0.1|66.1|66.0|64.6|
> > |0.2|67.0|67.7|66.3|
> > |0.5|68.8|69.5|68.5|
> > |1|68.8|68.7|68.4|
> > |2|68.5|68.3|41.8|
> > |5|67.4|67.6|34.2|
> > |10|63.8|63.8|29.7|
> >
> > **Table 8** MNIST, QDA accuracy vs regularization
> > |regularization|SQFA|SQFA-H|SQFA-B|
> > |-------------:|---:|-----:|-----:|
> > |0.05|81.2|92.7|89.1|
> > |0.1|84.4|93.2|90.3|
> > |0.5|88.7|93.6|91.5|
> > |1|88.9|93.6|91.7|
> > |2|90.6|93.3|91.7|
> > |5|91.1|93.0|91.3|
> > |10|91.4|92.5|73.7|
> > |20|91.4|91.1|70.5|
> > |50|86.7|86.4|62.9|
> >
> > >**It would be great to provide more explicit guidance on initialization, convergence behavior, and runtime scaling in practice.**
> >
> > Regarding initialization and convergence, we write in the Methods that filters are initialized to random unit vectors, and that “the performance across 20 different initializations was very similar.”. So similar were the results of different initializations, that the error bars are barely visible. For example, for the SVHN dataset, the 20 randomly initialized SQFA repeats have a median QDA accuracy of 68.8%, and the 95% quantile is [67.9%, 68.9%]. Thus, initialization did not require finetuning, and convergence was remarkably consistent. We will provide this quantitative convergence data in the supplementary materials, and emphasize it in the Methods section.
> >
> > Regarding runtime scaling, we tested the method on toy datasets with much larger data dimensionalities (up to $2\times10^4$) and learned filters (up to $500$) in response to reviewer CMDY (**Table 2,3**). Optimization was still completed in the order of seconds in a consumer laptop. We will include this data in the revised paper.
> >
> > >**Consider moving a few key points from Appendix B/C/H into the main paper, e.g., (i) the main intuition from Appendix B for the zero-mean case, (ii) a compact quantitative check from Appendix C showing how faithful the Calvo–Oller surrogate is in practice, and (iii) the main takeaway from Appendix H on what happens when Gaussianity is violated.**
> >
> > We thank the reviewer for going over the Appendix and making these suggestions.
> >
> > The main intuition from Appendix B can be unpacked where we make reference to the appendix, right after presenting the formula for the FR distance in the zero-mean case.
> >
> > We agree that a compact quantitative check of the Calvo-Oller behavior in practice would nicely fit the main text. We will report for each dataset summary statistics describing how much the Calvo-Oller bound deviates from the Fisher-Rao distance.
> >
> > Regarding the violation of Gaussianity, a key point there is that the SQFA methods will learn the exact same filters whether the data is Gaussian or not, because they only use summary statistics (similar to how LDA behaves). We will emphasize this point in the main text. We can mention it in the Methods Optimization section, after describing that optimization is based on the conditional class statistics alone, and mention it again in the Discussion. Then, the key takeaway–that the results are similar when either using the kNN classifier for evaluation, or testing on Gaussian data with statistics matched to the real datasets–can be mentioned in the edits related to the kNN classifier requested by the reviewer in a comment above.

---

> > ### Comment · Reviewer_bDFH · 2026-04-21
> > **Please include results in main paper**
> >
> > Please surface these results up to the main paper. The full tables can stay in an appendix, but I think at least one compact plot (accuracy vs #filters) or a small summary table should be in the main text. That will strengthen the paper by a lot.
> >
> > According to the table, the KNN accuracy for speed-estimation drops sharply as the number of filters increases (including for SQFA variants). That might be a KNN/metric its own issue like curse of dimensionality, choice of k, etc., but if we want to use KNN as evidence beyond QDA, it would help to briefly explain this behavior and/or include a more stable non-QDA decoder for that dataset (e.g., logistic regression / linear SVM) to avoid confusing messages.

---

> > > ### Author Response · Authors · 2026-04-30
> > >
> > > > **Please surface these results up to the main paper. The full tables can stay in an appendix, but I think at least one compact plot (accuracy vs #filters) or a small summary table should be in the main text. That will strengthen the paper by a lot.**
> > >
> > > We agree that this will considerably strengthen the paper, and should be high priority to include in the main paper. Thank you for the original suggestion.
> > >
> > > > **According to the table, the KNN accuracy for speed-estimation drops sharply as the number of filters increases (including for SQFA variants). That might be a KNN/metric its own issue like curse of dimensionality, choice of k, etc., but if we want to use KNN as evidence beyond QDA, it would help to briefly explain this behavior and/or include a more stable non-QDA decoder for that dataset (e.g., logistic regression / linear SVM) to avoid confusing messages.**
> > >
> > > We agree that this somewhat unexpected behavior should be addressed in the final manuscript. As the reviewer mentions, it is likely a curse of dimensionality effect impacting this specific dataset and classifier combination. We will not have time to address this within the remaining rebuttal window, but it should not be difficult to pinpoint the issue for the final revisions if the paper is accepted.

---

> ### Comment · Reviewer_bDFH · 2026-04-15
> **Non-QDA eval / “closed loop”**
>
> * Non-QDA eval
>
> Thank you. I missed Appendix H. And yes, I think it's necessary to surface some results in H up to the main paper.
>
> *  “closed loop”
>
> QDA is not necessarily an unfair eval. My point was that if the goal is “features that work well for QDA,” then evaluating with QDA makes sense. My concern is more about what a reader can conclude beyond that. A lightweight non-QDA check (KNN is totally fine) helps show the learned features aren’t only useful under the exact same Gaussian decoder assumptions.

---

> ### Author Response · Authors · 2026-05-01
>
> We agree with the framing proposed. Implying that Fisher-Rao is expected to be superior was not our aim, as we tried to make  explicit by saying that other distances have a more direct relation to Bayes error. In the revised manuscript we will make the framing clearer earlier.
>
> Then, in practical terms, we show that the Fisher-Rao distance maximization leads to competitive performance, better or on par with every other method (except the Hellinger distance). This is a first step showing that it is possible to maximize Fisher-Rao distances in this scenario, and that the intuitive information geometry of the problem translates into a useful learning objective. This is motivated by the intrinsic geometric and scientific appeal of the Fisher-Rao distance (e.g. the Bhattacharyya distance is not a distance, and the Hellinger distance is not a geodesic distance). But then, while we could speculate about other practical reasons why the Fisher-Rao distance might be preferred (e.g. robustness), these would have to be tested empirically, which is something to be done in future work. We can add a paragraph in the discussion about this question, and pointing it out as a venue of future work. But the main reasons at this time for using the Fisher-Rao distance are that we show it to be competitive across problems and conditions, and the interest that information geometry itself draws as an emerging tool in different areas.

---

### Review · Reviewer_hHCx · 2026-04-04

**Summary Of Contributions:**

The paper proposes Supervised Quadratic Feature Analysis (SQFA), a linear supervised dimensionality-reduction method that learns filters by maximizing Fisher–Rao distances between class-conditional Gaussians in the low-dimensional feature space. To instantiate the idea, the authors analyze both zero-mean special cases and also apply the Calvo–Oller (CO) affine-invariant lower bound for general cases. They demonstrate the usefulness of SQFA on a synthetic toy case and simple image classification tasks.

**Audience:**

No

**Audience Explanation:**

This work proposes to adopt Fisher–Rao distance as the discriminability proxy, and discuss two direct instantiations. It's unclear how the two cases align with practical applications. And the empirical results do not sufficiently justify the advantages of Fisher–Rao distance .

**Claims And Evidence:**

No

**Claims Explanation:**

The authors argue that Fisher–Rao distance is a principled proxy for discriminability because it integrates local Fisher information along geodesics, or more suitable for classification. However, the experiment only provides qualitative studies, without large scale quantative evidence.

**Requested Changes:**

1. It's suggested to list the contributions and the key technical novelty of this work at the end of the introduction.

2. It's suggested to provide more discussion on how the two cases align with the realistic applications, e.g., whether the assumptions hold; whether optimizing the lower bound could increase the true Fisher–Rao distance; what's the advantage of increased Fisher–Rao distance;

3. The benchmarks cover only samll scale and limited cases. It's unclear whether this method will also be effective in more realistic and large-scale benchmarks.

---

> ### Author Response · Authors · 2026-04-28
> **Key contributions, discuss assumptions**
>
> > **It's suggested to list the contributions and the key technical novelty of this work at the end of the introduction.**
>
>
> We will include something like the following text at the end of the introduction.
>
> > *In this work, we show that:*
> > - *For dimensionality reduction, maximizing Fisher-Rao distances between classes leads to competitive classification accuracy.*
> > - *When directly maximizing Fisher-Rao distances is computationally impractical, the Calvo-Oller bound is a good substitute.*
> > - *Maximizing the Hellinger distance consistently outperforms other distances and information theoretic quantities for accuracy maximization, despite the fact that it is a rarely used optimization objective.*
> >
> > *These results provide novel insights into the uses of Fisher-Rao distances for machine learning, and a new state of the art method for learning features that maximize the accuracy of quadratic decoders.*
>
> > **It's suggested to provide more discussion on how the two cases align with the realistic applications, e.g., whether the assumptions hold**
>
>
> We thank the reviewer for raising these points. The zero-mean assumption is appropriate for several applications, although certainly not all. Some examples are some types of neural data (e.g. EEG data), local visual tasks (e.g. motion estimation, disparity estimation, focus error estimation), and auditory waveform data.
>
> In our paper, the main role of the zero-mean case is that it helps understand Fisher-Rao distances as a learning objective, since it provides us with a closed-form expression. See the second comment, last point, in response to CMDY. We will clarify this in the updated manuscript.
>
> Regarding the assumption of Gaussianity, it is a standard assumption in many linear dimensionality reduction methods. This assumption is often violated, but it also applies to good approximation in many cases. Non-parametric dimensionality reduction methods (e.g. WDA, LMNN, LFDA) can be useful for highly non-Gaussian datasets, but these methods have their own shortcomings, making Gaussian methods still a popular choice among practitioners. Furthermore, SQFA variants outperformed non-parametric methods in the datasets used in this work, which are not Gaussian. This shows that SQFA has some degree of robustness to deviations from Gaussianity (and see new experiments in **Tables 4,5**).
>
> Finally, future work can extend the method to non-Gaussian elliptical distributions, as mentioned in the Discussion. We will further discuss these points in an updated manuscript.
>
> > **It's suggested to provide more discussion on how the two cases align with the realistic applications, e.g. … whether optimizing the [Calvo-Oller] lower bound could increase the true Fisher–Rao distance;**
>
> This is a good point. Appendix C1 of the submitted paper shows that this bound is a good approximation to the Fisher-Rao distance for the datasets used. A summary of these results will be moved to the main text.
>
> To further support the use of the Calvo-Oller (CO) bound, we performed a new analysis. We numerically computed the true average Fisher-Rao distance between classes in the feature spaces learned with a range of different objective functions (see Table 1-3), including CO bound maximization. If CO bound maximization increases the Fisher-Rao distance, we expect it to have the highest average Fisher-Rao distance.
>
> Tables 1-3 show the mean Fisher-Rao distance for different learning objectives (columns) and different numbers of filters (rows). Across all conditions, CO bound maximization led to the highest average Fisher-Rao distance. This supports the use of CO bound maximization as a practical way to maximize Fisher-Rao distances in learning. We will include these results in the updated manuscript.
>
> **Table 1. Mean Fisher-Rao distance, SVHN**
> | N filters | Calvo-Oller | Hellinger | Bhattacharyya | Wasserstein | Jeffreys | PCA | LDA |
> | --| --| --| --| --| --| --| --|
> | 2 | **1.42** | 1.20 | 1.35 | 1.20 |1.36 |0.12 | 0.37 |
> | 4 | **2.31** | 2.24 | 2.29 | 1.87 |2.27 | 0.55 | 0.92 |
> | 8 | **2.51** | **2.51** | 2.45 | 2.12 |2.47 | 1.13 | 1.07 |
> |16 | **3.50** | 3.49 | 3.49 | 3.05 |3.44 | 2.26 | |
>
>
> **Table 2. Mean Fisher-Rao distance, MNIST**
> | N filters | Calvo-Oller | Hellinger | Bhattacharyya | Wasserstein | Jeffreys | PCA | LDA |
> | --| --| --| --| --| --| --| --|
> | 2 | **2.92** | 2.86 | 2.80 | 2.66 |2.73 | 2.40 | 2.22 |
> | 4 | **3.90** | 3.79 | 3.84 |3.54 |2.77 | 3.21 | 3.16 |
> | 8 | **3.85** |3.49 | 3.56 | 3.66 |3.76 | 3.56 | 3.02 |
> |16 | **4.62** | 4.31 | 4.55 | 4.40 |4.42 | 4.34 | |
>
>
> **Table 3. Mean Fisher-Rao distance, Speed estimation**
> | N filters | Calvo-Oller | Hellinger | Bhattacharyya | Wasserstein | Jeffreys | PCA | LDA |
> | --| --| --| --| --| --| --| --|
> | 2 | **3.34** | **3.34** | **3.34** | 2.97 | **3.34** | 2.81 | 1.30 |
> | 4 | **4.33** | 4.27 |4.32 | 3.97 |3.91 | 3.72 | 3.09 |
> | 8 | **5.36** |5.19 | 5.27 | 5.09 |4.51 | 4.86 |5.26 |

---

> ### Author Response · Authors · 2026-04-29
> **Scale of the datasets, example 1**
>
> > **The benchmarks cover only small scale and limited cases. It's unclear whether this method will also be effective in more realistic and large-scale benchmarks.**
>
> Regarding the scale of the problems, we performed a scaling analysis with a toy problem in response to reviewer CMDY, learning hundreds of filters in up to tens of thousands of dimensions. See **Tables 2-3** in our response to them.
>
> We also extended our analysis of the current benchmarks by testing how performance changes with the number of filters, and by including the new comparison methods SQFA-W (for Wasserstein distance), SQFA-J (for Jeffreys divergence), Local Fisher Discriminant Analysis (LFDA) and Wasserstein Discriminant Analysis (WDA) in response to the other two reviewers.
>
> However, it is not clear to us what type of large-scale realistic benchmark the reviewer has in mind that would fit the scope of the present paper. The goal of SQFA is to learn linear features useful for the QDA classifier. Supervised linear dimensionality reduction methods like SQFA are rarely applied to raw large-scale benchmarks such as Imagenet. This is for good reason. If the goal is to maximize classification performance, linear supervised dimensionality reduction followed by a simple classifier would not achieve competitive performance with state of the art methods in large-scale datasets. If the goal is to reduce the dimensionality  for pre-processing, often PCA will be preferred in large-scale datasets due to its low computational complexity. Linear supervised dimensionality reduction methods are often used on smaller-scale datasets to extract insights, when the data is limited, or when there is a priori reason to believe that linear dimensionality reduction is well-suited to a particular problem. As such, the benchmarks used in this work are in line with the linear dimensionality reduction literature in general (e.g. see Flamary 2018, Weinberger 2009).
>
> With this in mind, we tested our method on two additional datasets that reflect  realistic problems.
> First, to reflect an analysis of a larger more complex dataset, we used the dataset CIFAR100, consisting of 100 classes with 600 images each. We first computed the 512 dimensional embeddings of the dataset images using a pretrained ResNet-18, to obtain a representation more suitable for supervised linear dimensionality reduction. Then we applied dimensionality reduction to these embeddings as done for the datasets in the main text. The results show that for this larger scale dataset, accuracy with SQFA features remains competitive, outperforming most other methods, and that SQFA-H leads to the highest accuracy for all numbers of filters learned  (**Table 4**).
>
> **Table 4: QDA accuracy (%) for CIFAR100 embeddings (two highest accuracies in bold)**
> | N filters |SQFA | SQFA-H | SQFA-B | SQFA-W | SQFA-J | LDA |SPCA |LFDA | WDA |LMNN | PCA |
> | - | - | -- | -- | -- | -- | - | - | - | - | - | - |
> | 2 | **10.8** |**11.2** |10.3 |8.1 |9.1 | 11.1 | 8.0 | 10.4 |8.3 |9.3 |7.0 |
> | 4 | 20.0 |**24.5** | **21.7** |17.7 |19.2 | 20.1 | 17.6 | 19.5 |16.4 | 21.0 | 16.4 |
> | 8 |32.6 | **40.4** |34.2 |32.1 |32.0 |33.1 | 32.2 | 32.1 | 28.0 | **35.7** | 29.4 |
> |16 | 45.9 | **50.2** |47.4 |45.9 |45.3 | 45.4 | 45.5 | 42.9 | 41.6 | **48.8** | 43.9 |

---

> ### Author Response · Authors · 2026-04-29
> **Scale of the datasets, example 2**
>
> Then, we analyzed a real-world scientific problem for which SQFA can be useful: dimensionality reduction of neural recordings. An important question in neuroscience is how condition-dependent changes in neural response covariance affect neural coding [Kohn 2016]. In neural recordings it is common to have many neurons but few trials per condition. In such cases, finding subspaces of neural activity that capture changes in the response covariance and that are relevant for decoding experimental condition, can be specially useful to study the effects of condition-dependent covariability on neural coding.
>
> For our second analysis, we used an openly available dataset [Henry & Kohn 2022], where the authors record the response of neurons in primary visual cortex of macaque monkey to 4 different visual stimuli, where an aim is to quantify the effects of response variability on neural coding. The responses are recorded as spike counts (positive integers).
>
> This dataset consists of 22 experimental sessions, each with one to two dozen recorded neurons, and 150-200 total trials (37 to 50 trials per condition). The neural responses are noisy and highly non-Gaussian. To evaluate the different dimensionality reduction models, we did 10 independent train/test splits for each of the 22 experimental sessions, and for each one we learned features with the different methods on the training set, and evaluated QDA accuracy on the testing set. This procedure resulted in 220 fits and evaluations per method. We learned 2 filters, since the dataset consists of only 4 classes and one to two dozen dimensions.
>
> We observe that the average accuracy achieved with SQFA filters is second only to SQFA-H filters (**Table 5**), in line with the results for the rest of the datasets.
>
> **Table 5: QDA accuracy (%) for primary visual cortex recordings (two highest accuracies in bold)**
> | N filters |SQFA | SQFA-H | SQFA-B | SQFA-W | SQFA-J | LDA |SPCA |LFDA | WDA |LMNN | PCA |
> | - | - | -- | -- | -- | -- | - | - | - | - | - | - |
> | 2 | **33.8** |**34.2** |33.2 |32.6 |33.2 | 33.2 | 32.6 | 32.7 | 29.4 | 29.4 | 27.3 |
>
> These two datasets provide additional diverse examples, showing that maximizing Fisher-Rao distances leads to competitive classification accuracy, and that maximizing the Hellinger distance leads to the highest performance across methods and conditions. We will include these datasets in the revised submission.
>
> **References**
>
> Flamary, R., Cuturi, M., Courty, N., & Rakotomamonjy, A. (2018). Wasserstein discriminant analysis. *Machine Learning*, 107(12), 1923-1945.
>
> Weinberger, K. Q., & Saul, L. K. (2009). Distance metric learning for large margin nearest neighbor classification. *Journal of machine learning research*, 10(2).
>
> Kohn, Adam, et al. "Correlations and neuronal population information." *Annual review of neuroscience* 39.1 (2016): 237-256.
>
> Henry, C. A., & Kohn, A. (2022). Feature representation under crowding in macaque V1 and V4 neuronal populations. *Current Biology*, 32(23), 5126-5137.

---

> ### Author Response · Authors · 2026-04-30
> **Additional Calvo-Oller bound results**
>
> Additionally, we also repeated the analysis of computing the true average Fisher-Rao distances between classes in the different learned feature spaces for the new datasets included above. Again, maximizing the Calvo-Oller bound led to the highest Fisher-Rao distances among optimization objectives (**Table 6,7**).
>
> **Table 6. Mean Fisher-Rao distance for first recording session of Henry 2022 dataset**
> | N filters | Calvo-Oller  | Hellinger | Bhattacharyya | Wasserstein | Jeffreys | PCA   | LDA   |
> | ---| --| --| --| --| --| --| ----- |
> | 2         | **2.49** | 2.13     | 2.46         | 1.51       | 2.48    | 0.56 | 1.43 |
>
> **Table 7. Mean Fisher-Rao distance for CIFAR100 embeddings**
> | N filters | Calvo-Oller  | Hellinger | Bhattacharyya | Wasserstein | Jeffreys | PCA   | LDA   |
> | --------- | ----- | --------- | ------------- | ----------- | -------- | ----- | ----- |
> | 2         | **2.74** | 2.71     | 2.67         | 2.26       | 2.55    | 1.91 | 2.65 |
> | 4         | **3.02** | 2.60     | 2.63         | 2.81       | 2.99    | 2.69 | 2.57 |
> | 8         | **2.81** | 2.15     | 2.72         |             |          |       |       |
>
> Note: Computing the Fisher-Rao distances numerically is computationally intensive. The CIFAR100 dataset requires computing many distances because of its high number of classes, so not all results have finished running yet. We will update the table as new results are obtained.

---

> > ### Comment · Reviewer_hHCx · 2026-05-08
> >
> > Thank you so much for providing the comprehensive new experimental results and discussions. They addressed my questions very well.

---

### Author Response · Authors · 2026-05-01
**Summary of the response to reviewers**

We thank the reviewers for their constructive comments that helped strengthen our work.

The comments led us to perform additional analyses which strengthened the article and further supported our main points that: i) maximizing the Fisher-Rao distance is a good proxy for discriminability in supervised dimensionality reduction, and ii) SQFA–and in particular SQFA-H–is a state-of-the-art method for learning features that support quadratic classification.

Specifically, we compared more methods (SQFA-W, SQFA-J, WDA and LFDA), used new datasets (CIFAR100, neural spiking dataset), and performed more robustness checks (accuracy vs number of filters).  The results further support the claims of the initial submission. If our article is accepted at TMLR, we will include these new analyses in the revised manuscript.

Additionally, in the responses we showed that Calvo-Oller bound maximization leads to the largest average Fisher-Rao distances among methods, that the model can be easily scaled to hundreds of filters and tens of thousands of dimensions, and that the learned features are useful for classifiers other than QDA.

In response to other comments, we also have clarified and discussed points like affine invariance, the differences between different distances, the Gaussianity assumptions, and other related methods. These extended discussion points will also be included in a revised manuscript.

---

### Decision · Action_Editor_ZodF · 2026-05-08

**Recommendation:** Accept with minor revision

**Additional Comments:**

The rebuttal produced several new results which should appear in the paper, and clarified that some aspects of the appendix should go to the main paper. I expect the authors to make these changes in the final copy.

Further, one reviewer expressed concerns about the motivation of the paper not being sufficiently concise. I agree that the paper should be improved by sharpening the argument as to *why* the Fisher-Rao metric is well-suited for discriminative dimensionality reduction. The current argument is based on a task-agnostic argument about the FR being a good metric.

**Audience:**

Yes

**Audience Explanation:**

The findings of the paper should be of interest to the information geometry community.

**Claims And Evidence:**

Yes

**Claims Explanation:**

All but one concern regarding claims was handled during the rebuttal period. One reviewer has a remaining concern about the paper potentially overclaiming on how applicable the proposed method might be in real-world problems. After having read the paper, I do not share this concern.

---

> ### Author Response · Authors · 2026-06-13
> **Camera ready version**
>
> We appreciate all the reviewers comments, that helped considerably strengthen the paper.
>
> We incorporated almost all suggested changes in the camera ready version. Of the two new datasets used in the rebuttal, we decided to incorporate only the Neural Recordings dataset, and not the CIFAR100 dataset, in order to keep the paper at a more manageable length. The results of that analysis will remain available at OpenReview, and we uploaded the code to reproduce that analysis to the paper's GitHub repository.